# Sketch-Based Low-Rank Model Merging with Shared Circulant Transforms

**Zhiming Zhang** [1 2 3]  **Rong Yin** [1 †]  **Xiaoshuai Hao** [4]  **Hansong Zhang** [2 3]
**Hao Peng** [1]  **Yong Liu** [5]  **Can Ma** [2 3]  **Dan Meng** [2 3]

## Abstract

Merging multiple low-rank adapters (LoRA) provides a practical route to scaling multi-task learning and deployment more efficiently than full-model weight merging, while avoiding task-specific training data. However, most existing approaches either treat LoRA updates as dense weight deltas or rely on expensive subspace factorizations, making merging a primary latency bottleneck. To address this issue, this paper first studies the TA objective as a tractable case and establishes a theoretically positive relationship between merging quality and the effective rank of the merged matrices. Motivated by this insight, we propose **CircuMerge**, a sketch-based framework for low-rank model merging built on shared circulant transforms. Specifically, it treats each adapter as a pair of low-rank matrices and applies a shared circulant transform to align all tasks in a common coordinate system. This alignment enables efficient sampling, allowing us to generate compact sketches that effectively summarize the interactions between tasks. These sketches enable applying merging rules directly to them and reconstructing a standard low-rank adapter, preserving essential information while significantly reducing computational overhead. Across broad multi-task LoRA benchmarks covering both vision and language settings, extensive empirical results demonstrate that CircuMerge reduces overall merging time by at least 44% compared to state-of-the-art approaches, while achieving optimal or near-optimal accuracy.

---

[1]School of Cyber Science and Technology, Beihang University, Beijing, China. [2]Institute of Information Engineering, Chinese Academy of Sciences, Beijing, China. [3]School of Cyber Security, University of Chinese Academy of Sciences, Beijing, China. [4]School of Computer Science and Technology, Hangzhou Dianzi University, Hangzhou, China. [5]Gaoling School of Artificial Intelligence, Renmin University of China, Beijing, China. Correspondence to: Rong Yin <yinrong@buaa.edu.cn>.

*Proceedings of the $43^{rd}$ International Conference on Machine Learning*, Seoul, South Korea. PMLR 306, 2026. Copyright 2026 by the author(s).

## 1. Introduction

The pre-training and fine-tuning paradigm is the standard route to high-performance specialized models: A large base model is pretrained on broad corpora (Devlin et al., 2019; Radford et al., 2019; 2021) and then adapted via supervised fine-tuning (Wortsman et al., 2022b). As base models scale, fully fine-tuning and storing a separate copy per task becomes increasingly costly, motivating parameter-efficient fine-tuning (PEFT) (Ding et al., 2023) methods such as LoRA (Hu et al., 2022), which inject low-rank adaptation modules into a frozen backbone instead of updating all weights, thereby substantially reducing training and storage costs. In parallel, open-source ecosystems (e.g., Hugging Face (Wolf et al., 2020)) have popularized the release of task-specific adapters alongside pretrained checkpoints, raising a practical question: How can we reuse and compose these assets without access to the original training data.

Zero-shot model merging (Wortsman et al., 2022a; Stoica et al., 2024; Huang et al., 2024) has attracted growing attention as a way to further reduce deployment and maintenance costs while allowing composition over many models, particularly through methods that merge task-specific parameter updates or adapter weights (Yadav et al., 2023; Ilharco et al., 2023; Zhang et al., 2023; Yu et al., 2024; Gargiulo et al., 2025; Yoshida et al., 2025) produced by fine-tuning.

In particular, Especially for LoRA-based adapters (Hu et al., 2022) attached to dense layers($l$) with weights $\mathbf{W}^{(1)} \in \mathbb{R}^{d_{\text{out}} \times d_{\text{in}}}$, low-rank parameterization $\Delta \mathbf{W}^{(1)} = \mathbf{A}^{(1)} \mathbf{B}^{(1)}$ with $\mathbf{A}^{(1)} \in \mathbb{R}^{d_{\text{out}} \times r}$ and $\mathbf{B}^{(1)} \in \mathbb{R}^{r \times d_{\text{in}}}$ for $r \ll \min(d_{\text{out}}, d_{\text{in}})$ suggests that merging between tasks could, in principle, be much more efficient than operating directly on full dense updates. However, most existing zero-shot task-specific parameter merging methods still materialize or optimize over dense task updates (Stoica et al., 2025; Zeng et al., 2025; Wei et al., 2025b), treating LoRA as generic PEFT residuals $\Delta \mathbf{W}^{(l)} \in \mathbb{R}^{d_{\text{out}} \times d_{\text{in}}}$ and leaving much of its structural advantage for merging untapped.

Recent work on core space merging exploits the low-rank structure of LoRA adapters by projecting task updates into a shared low-dimensional subspace before aggregation (Stoica et al., 2025; Panariello et al., 2025b). This line of work shows that merging in a low-dimensional space is

a promising way to leverage the LoRA-specific structure for effective and efficient merging; however, random sampling or projections in this space can discard important interactions between tasks, so (Stoica et al., 2025; Panariello et al., 2025b) still relies on relatively expensive matrix factorizations whose cost becomes a bottleneck when scaling to many tasks and layers.

To overcome these limitations, we develop an efficient, theory-guided approach that preserves task interactions under compressed low-rank merging. Specifically, we first study the Task Arithmetic (TA) (Ilharco et al., 2023) objective as a tractable case, where our analysis connects merging quality to spectrum uniformization: under mild conditions, a recoverable transformation that spreads singular energy, thereby increasing effective rank, leads to improved merging quality. This mechanism provides theoretical motivation for applying structured, invertible mixing in the LoRA rank space prior to sketching. Building on this perspective, we propose **CircuMerge**, which applies a shared invertible circulant transform in the LoRA rank space to place all tasks in a common coordinate system via structured mixing, and then constructs compact sketches by uniformly sampling a fixed set of rows and columns shared across tasks within each layer. This shared sampling yields consistent, comparable sketches across tasks while keeping the procedure lightweight and reproducible. We execute standard merging rules directly on the sampled submatrices in sketch space, and combine the remaining unsampled part by simple element-wise averaging across tasks. While our theoretical analysis is scoped to TA, we further evaluate **CircuMerge** as an empirical extension to other merging rules. Experiments on LoRA adapters fine-tuned from both vision and language pretrained models show that CircuMerge substantially reduces per-layer merging time while achieving optimal or near-optimal accuracy across diverse benchmarks.

In summary, this paper propose a sketch-based formulation for LoRA merging, which merges compact sketches without costly factorizations. Specifically, the main contributions are as follows:

- This paper proposes **CircuMerge**, a sketch-based LoRA merging framework that uses a shared circulant transform to enable structured, invertible rank-space mixing and effective sketch-space merging.

- For the Task Arithmetic objective, this paper establish a theoretical connection between merging quality and the effective rank of the matrices being merged.

- Experiments on LoRA adapters fine-tuned from both vision and language pretrained models show that CircuMerge reduces per-layer merging time by at least 44% while achieving optimal or near-optimal accuracy across diverse benchmarks.

## 2. Related Work

### 2.1. Model Merging

The rapid accumulation of expert checkpoints has made it increasingly important to understand how to consolidate multiple task specialists into a single model without relying on additional training data (Li et al., 2023; Yang et al., 2024a; Ruan et al., 2025). Early work investigates principled ensembling in weight space, such as Fisher-weighted averaging (Matena & Raffel, 2022). Building on this direction, a large body of work has explored data-free and zero-shot strategies for merging model weights (Jin et al., 2023; Huang et al., 2024; Stoica et al., 2024; Tam et al., 2024; Yang et al., 2024b; Du et al., 2024; Wei et al., 2025a; Panariello et al., 2025a; Akiba et al., 2025; Li et al., 2025), aiming to produce a unified model that retains as much expert performance as possible across diverse settings. Feature-level fusion studies, such as MapFusion (Hao et al., 2025), further show that effective fusion requires aligning heterogeneous representations rather than simply combining them.

A prominent line of work merges models in the space of weight updates which comes from fine-tuning to pretrained model, often represented as task vectors. Task Arithmetic (Ilharco et al., 2023) and related weight-space composition methods (Zhang et al., 2023) combine parameter deltas via simple arithmetic, while later approaches mitigate interference by resolving sign conflicts (Yadav et al., 2023), applying sparse masks to remove problematic updates (Davari & Belilovsky, 2024), or filtering destructive weights (Wang et al., 2024). More recent methods leverage structured representations of update matrices, including singular components and aligned subspaces, to improve multi-task composition, such as TSV (Gargiulo et al., 2025), ISO-C (Marczak et al., 2025), and KnOTS (Stoica et al., 2025).

An increasing body of work studies merging when the objects to be combined are parameter-efficient adapters (e.g., LoRA (Hu et al., 2022)) rather than full model weights. RobustMerge considers this setting in multimodal large models and emphasizes robustness of update directions under task interference (Zeng et al., 2025). Some LoRA-specific lines of work improve mergeability by imposing additional training-time structure, including FlyLoRA (Zou et al., 2025) and LORI (Zhang et al., 2025), as well as OSRM (Zhang & Zhou, 2025). While effective in their intended settings, these approaches rely on training-time choices and therefore do not directly address the more open setting of merging arbitrary off-the-shelf LoRA adapters released on public platforms. Another line of work combines LoRA with mixture-of-experts routing by treating multiple LoRA modules as experts. For example, MoLE (Wu et al., 2024) improves LoRA fusion through hierarchical gating over multiple LoRA components, while MeteoRA (Xu et al., 2025) embeds multiple task-specific LoRA adapters into a

base LLM via a full-mode MoE architecture for automatic adapter selection. LoRAMoE (Dou et al., 2024) further explore router-based LoRA expert composition for instruction tuning and parameter-efficient adaptation. Different from these routing-based approaches, our method compresses and merges LoRA adapters into a single standard low-rank adapter, avoiding additional router training and multi-expert inference overhead. Recent work also explores merge-time alignment into shared low-rank representations. Core Space Merging (Panariello et al., 2025b) constructs a shared basis and performs merging on compact core matrices whose size scales with the adapter rank, offering an efficiency-oriented perspective for low-rank adapter merging. These lines of work either rely on special training-time structures or require expensive matrix factorizations at merge time, which limits their broad applicability.

## 2.2. Matrix Sketching of Model Weights

Matrix sketching (Hinrichs & Vybíral, 2011; Yin et al., 2020a; 2022a;b) provides compact surrogates for large matrices while approximately preserving their action on input vectors. Prior neural-network applications, including SS1 (Saedi et al., 2024), and random-basis training (Gressmann et al., 2020), mainly use sketch-structured representations to reduce training or inference cost for individual weight operators. In contrast, we apply sketching at merge time to capture and combine cross-task interactions(Yin et al., 2021). To make this process efficient, we adopt circulant sketching, an FFT-friendly structured projection with fast multiplication costs (Yin et al., 2020b; 2023).

## 3. Preliminaries

### 3.1. Circulant Matrices

A circulant matrix is determined by its first column $\mathbf{c} = (c_0, c_1, \ldots, c_{n-1})^\top \in \mathbb{R}^n$, denoted by $\mathbf{C} = \mathrm{circ}(\mathbf{c}) \in \mathbb{R}^{n \times n}$. Concretely,

$$\mathbf{C} = \begin{bmatrix} c_0 & c_{n-1} & c_{n-2} & \cdots & c_1 \\ c_1 & c_0 & c_{n-1} & \ddots & \vdots \\ c_2 & c_1 & c_0 & \ddots & c_{n-2} \\ \vdots & \ddots & \ddots & \ddots & c_{n-1} \\ c_{n-1} & \cdots & c_2 & c_1 & c_0 \end{bmatrix}, \quad (1)$$

with entries

$$\mathbf{C}_{i,j} = \mathbf{c}_{(i-j) \bmod n}, \qquad i, j \in \{0, \ldots, n-1\}. \quad (2)$$

In the proposed method, the circulant transform is initialized following a standard construction from the structured random projection literature. Specifically, we first sample a random vector $\widetilde{\mathbf{c}} = (\widetilde{c}_0, \widetilde{c}_1, \ldots, \widetilde{c}_{n-1})^\top$ with entries

$\widetilde{c}_i \sim \mathcal{N}(0, 1)$, and then normalize it to obtain

$$\mathbf{c} = \frac{\widetilde{\mathbf{c}}}{\|\widetilde{\mathbf{c}}\|_2}. \quad (3)$$

The corresponding circulant matrix $\mathbf{C} = \mathrm{circ}(\mathbf{c})$ is used as the shared structured transform and is kept fixed during merging. Thus, the transform introduces no additional training cost, while retaining the storage and computational advantages of circulant matrices.

**Diagonalization via Discrete Fourier Transform (DFT).** Let $\mathbf{F} \in \mathbb{C}^{n \times n}$ be the unitary DFT matrix ($\mathbf{F}^* \mathbf{F} = \mathbf{I}$). Circulant matrices are diagonalized by the DFT:

$$\mathbf{C} = \mathbf{F}^* \mathrm{diag}(\widehat{\mathbf{c}}) \, \mathbf{F}, \qquad \widehat{\mathbf{c}} := \mathbf{F}\mathbf{c}. \quad (4)$$

Specifically, $\mathbf{C}$ is invertible iff all $\widehat{c}_k \neq 0$, in which case $\mathbf{C}^{-1}$ is obtained by elementwise inversion of $\widehat{\mathbf{c}}$ in the diagonal.

### 3.2. Fast Multiplication via Fast Fourier Transform

Multiplying a circulant matrix by a vector corresponds to circular convolution. For any $\mathbf{x} \in \mathbb{R}^n$,

$$\mathbf{C}\mathbf{x} = \mathbf{F}^* \Big( (\mathbf{F}\mathbf{c}) \odot (\mathbf{F}\mathbf{x}) \Big), \quad (5)$$

where $\odot$ denotes elementwise multiplication. Thus $\mathbf{C}\mathbf{x}$ can be computed using two Fast Fourier Transform (FFTs) and one inverse FFT in $O(n \log n)$ time (versus $O(n^2)$ for dense multiplication). Similarly, applying $\mathbf{C}^{-1}$ replaces $(\mathbf{F}\mathbf{c})$ by $(\mathbf{F}\mathbf{c})^{-1}$ (elementwise), assuming $\widehat{\mathbf{c}}$ has no zeros (and is not too close to zero for numerical stability).

For matrix inputs such as $\mathbf{A}\mathbf{C}$ with $\mathbf{A} \in \mathbb{R}^{d \times n}$ or $\mathbf{C}^{-1}\mathbf{B}$ with $\mathbf{B} \in \mathbb{R}^{n \times d}$, the same FFT procedure is applied independently to each row/column depending on the multiplication side, yielding a batchable $O(d \, n \log n)$ complexity rather than a dense $O(d \, n^2)$ cost.

## 4. Theory: Spectral Uniformization Improves Subsection Alignment Ratio

This section establishes a theoretically positive relationship between merging quality and the effective rank of the matrices being merged. Intuitively, effective rank captures how broadly the singular-value energy is spread across independent directions. When the merged matrix has low effective rank, its energy concentrates on a few dominant modes, so merge-time or post-merge compression tends to preserve only those modes and discard task-specific directions, degrading quality. When the effective rank is higher, the energy is distributed more evenly, yielding a higher-dimensional principal subspace at a fixed energy threshold and thus retaining a richer set of task-relevant directions.

Since effective rank is computed directly from singular values, this mechanism is measurable and can be checked layer-wise in practice.

## 4.1. Definitions

Fix one layer and let task ($t$) updates be $\{\Delta^{(t)}\}_{t=1}^{T} \subset \mathbb{R}^{d_{\text{out}} \times d_{\text{in}}}$. TA forms

$$M \triangleq \Delta^{\text{TA}} = \frac{1}{T} \sum_{t=1}^{T} \Delta^{(t)}. \tag{6}$$

Consider a shared invertible left perturbation $D \in \mathbb{R}^{d_{\text{out}} \times d_{\text{out}}}$ applied to all tasks. Then TA commutes with $D$:

$$\widetilde{M} \triangleq \frac{1}{T} \sum_{t=1}^{T} D\Delta^{(t)} = DM. \tag{7}$$

Given a reference matrix $X$, let $\Pi_{k,X}$ be the orthogonal projector onto the top-$k$ left singular subspace of $X$. For $\varepsilon \in (0, 1)$, define the $\varepsilon$-energy rank

$$k_\varepsilon(X) \triangleq \min\left\{k : \sum_{i \leq k} \sigma_i^2(X) \geq (1-\varepsilon)\|X\|_F^2\right\}. \tag{8}$$

We adopt the *Subsection Alignment Ratio* (SAR) introduced in (Marczak et al., 2025) to quantify how well a reference matrix $X$ preserves the task-relevant *output-direction subspace* of a target update $\Delta$. Let the (thin) SVD of $X \in \mathbb{R}^{d_{\text{out}} \times d_{\text{in}}}$ be $X = U\Sigma V^\top$. For any $k \in [d_{\text{out}}]$, let $U_k(X) \in \mathbb{R}^{d_{\text{out}} \times k}$ denote the top-$k$ left singular vectors of $X$, and define the orthogonal projector onto the corresponding principal subspace:

$$\Pi_{k,X} \triangleq U_k(X)U_k(X)^\top \in \mathbb{R}^{d_{\text{out}} \times d_{\text{out}}}. \tag{9}$$

Then $\Pi_{k,X}\Delta$ is the component of $\Delta$ aligned with the principal left-singular subspace induced by $X$. SAR measures the fraction of $\Delta$'s energy retained in this subspace:

$$\text{SAR}_k(\Delta, X) \triangleq \frac{\|\Pi_{k,X}\Delta\|_F}{\|\Delta\|_F} \in [0, 1], \tag{10}$$

where $\|\cdot\|_F$ is the Frobenius norm.

To make the subspace dimension comparable across layers/settings with different spectral concentration of $X$, we use the induced (adaptive) SAR by choosing $k$ as the $\varepsilon$-energy rank $k_\varepsilon(X)$ (Eq. (8)):

$$\text{SAR}_\varepsilon(\Delta, X) \triangleq \frac{\|\Pi_{k_\varepsilon(X),X}\Delta\|_F}{\|\Delta\|_F}. \tag{11}$$

To formalize "more uniform singular-value energy", we use the participation-ratio effective rank

$$r_{\text{eff}}(X) \triangleq \frac{\left(\sum_i \sigma_i^2(X)\right)^2}{\sum_i \sigma_i^4(X)} = \frac{1}{\sum_i p_i^2(X)}. \tag{12}$$

Larger $r_{\text{eff}}(X)$ means that the singular energy is less concentrated on a few dominant directions.

## 4.2. Uniformizing the Spectrum Increases Expected SAR

The remaining step is to connect $k_\varepsilon(X)$ to *expected* SAR. We adopt a standard isotropy model for typical task updates, motivated by the fact that a shared invertible mixing can reduce directional bias and make task updates approximately isotropic relative to the TA principal subspace.

**Assumption 4.1** (Left-isotropic task updates). A random task update $\Delta$ satisfies $\mathbb{E}[\Delta\Delta^\top] = \alpha I_{d_{\text{out}}}$ for some $\alpha > 0$, and is independent of the TA-induced principal subspaces.

**Theorem 4.2** (Spectral uniformization improves expected SAR (TA case)). *Let $M = \Delta^{\text{TA}}$ and $\widetilde{M} = DM$, where $D$ is invertible. Under Assumption 4.1, it holds that*

$$\mathbb{E}\left[\text{SAR}_\varepsilon^2(\Delta, \widetilde{M})\right] - \mathbb{E}\left[\text{SAR}_\varepsilon^2(\Delta, M)\right] = \frac{k_\varepsilon(\widetilde{M}) - k_\varepsilon(M)}{d_{\text{out}}}. \tag{13}$$

*Moreover, the expected squared SAR admits the following effective-rank lower bounds:*

$$\mathbb{E}\left[\text{SAR}_\varepsilon^2(\Delta, \widetilde{M})\right] \geq \frac{(1-\varepsilon)^2}{d_{out}} r_{\text{eff}}(\widetilde{M}), \tag{14}$$

*and*

$$\mathbb{E}\left[\text{SAR}_\varepsilon^2(\Delta, M)\right] \geq \frac{(1-\varepsilon)^2}{d_{out}} r_{\text{eff}}(M). \tag{15}$$

*In particular, if the shared invertible perturbation increases spectral uniformity ($r_{\text{eff}}(\widetilde{M}) > r_{\text{eff}}(M)$) and the increase is large enough to raise the $\varepsilon$-energy rank (e.g., $(1-\varepsilon)^2 r_{\text{eff}}(\widetilde{M}) > k_\varepsilon(M)$), then*

$$\mathbb{E}\left[\text{SAR}_\varepsilon^2(\Delta, \widetilde{M})\right] > \mathbb{E}\left[\text{SAR}_\varepsilon^2(\Delta, M)\right].$$

*Proof.* The proof is deferred to Appendix Section C.4. □

## 4.3. Methodological Implications and Motivation

Theorem 4.2 identifies a verifiable pathway to improved merging quality: a shared invertible reparameterization $D$ that makes the singular-value energy of $DM$ more uniform (higher $r_{\text{eff}}(DM)$) enlarges the principal subspace and increases the expected SAR. Building on this insight, we instantiate $D$ with a *Shared Circulant Transform* (SCT), motivated by structured Johnson–Lindenstrauss embeddings based on (preconditioned) circulant matrices and subsampling, which admit fast FFT-based application (Krahmer et al., 2014). In this work, SCT is a square circulant operator shared across tasks, and is invertible by construction via its Fourier-domain parametrization (all spectral coefficients

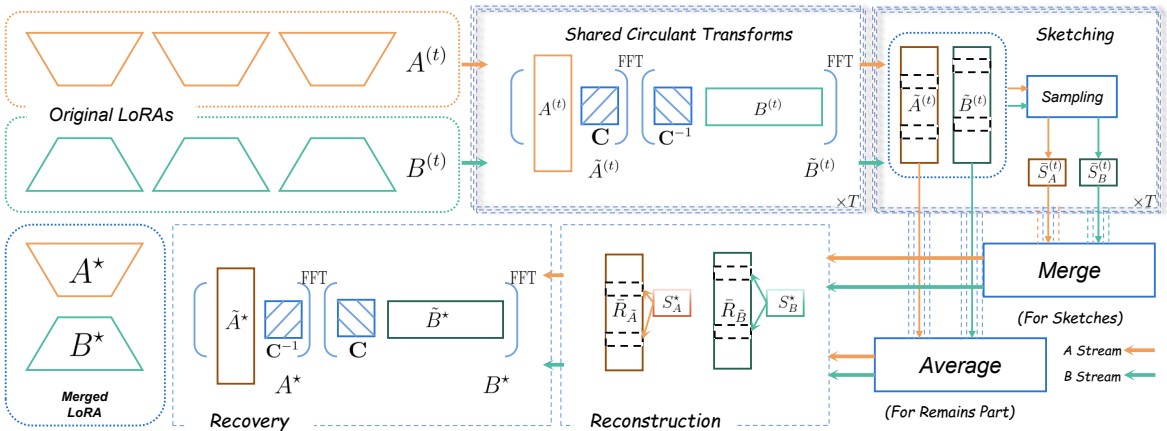

*Figure 1.* Overall pipeline of the proposed Approach CircuMerge. CircuMerge aligns task-specific LoRA factors $\{A^{(t)}, B^{(t)}\}$ using a shared circulant rank-space transform implemented with the *fast Fourier transform (FFT)*. It then builds compact row/column sketches via sampling, applies merging rules directly on sketches (averaging the unsketched remainder), and finally reconstructs and inverts the shared transform to obtain the merged LoRA $(A^\star, B^\star)$.

are bounded away from zero). We then apply random sampling to form compact sketches, and empirically verify that SCT increases $r_{\text{eff}}(\mathbf{DM})$ and flattens the spectrum, aligning with the mechanism of Theorem 4.2.

## 5. The Proposed Approach: CircuMerge

### 5.1. Problem Setting

We are given a frozen backbone with a set of LoRA-instrumented layers. For a specific layer with base weights $W_0 \in \mathbb{R}^{d \times d}$, each task $t \in \{1, \ldots, T\}$ is associated with a LoRA update

$$\Delta \boldsymbol{W}^{(t)} = \boldsymbol{A}^{(t)} \boldsymbol{B}^{(t)}, \quad \boldsymbol{A}^{(t)} \in \mathbb{R}^{d \times r}, \ \boldsymbol{B}^{(t)} \in \mathbb{R}^{r \times d}. \tag{16}$$

(For simplicity, we absorb the scaling factor into $A^{(t)}$ or $B^{(t)}$.) The goal of this problem is to construct a merged LoRA $(A^\star, B^\star)$ such that

$$\boldsymbol{W}^\star = \boldsymbol{W}_0 + \boldsymbol{A}^\star \boldsymbol{B}^\star \tag{17}$$

performs competitively across all tasks $\{1, \ldots, T\}$. We next describe the proposed circulant-reparameterized, sketch-based merging pipeline.

### 5.2. CircuMerge

Guided by the theoretical insights and spectral observations discussed above in Section 4.3, we propose **CircuMerge**, an efficient fusion framework depicted in Figure 1, focusing on the synergy between structural fidelity and efficiency.

**Shared Circulant Transforms.** For each LoRA-instrumented layer, we fix an invertible circulant matrix $C \in \mathbb{R}^{r \times r}$ that is shared across all tasks (see Section 3.1).

We reparameterize each task-specific LoRA update as

$$\begin{aligned} \Delta \boldsymbol{W}^{(t)} &= \boldsymbol{A}^{(t)} \boldsymbol{B}^{(t)} \\ &= \big(\boldsymbol{A}^{(t)} \boldsymbol{C}\big)\big(\boldsymbol{C}^{-1} \boldsymbol{B}^{(t)}\big) \qquad (18) \\ &\triangleq \tilde{\boldsymbol{A}}^{(t)} \tilde{\boldsymbol{B}}^{(t)}, \end{aligned}$$

where

$$\tilde{\boldsymbol{A}}^{(t)} = \boldsymbol{A}^{(t)} \boldsymbol{C} \in \mathbb{R}^{d \times r}, \quad \tilde{\boldsymbol{B}}^{(t)} = \boldsymbol{C}^{-1} \boldsymbol{B}^{(t)} \in \mathbb{R}^{r \times d}. \tag{19}$$

This keeps $\Delta \boldsymbol{W}^{(t)}$ unchanged while expressing all tasks in a shared (rank-space) coordinate system induced by $C$; computing $C^{-1}$ follows the standard circulant-matrix construction (see Section 3.1), and multiplications by $C$ (and $C^{-1}$) can be accelerated via FFT-based fast circulant multiplication (see Section 3.2).

**Sketching.** Let $s \ll d$ be a sketch size. We choose index sets $\boldsymbol{I}, \boldsymbol{J} \subseteq \{1, \ldots, d\}$ with $|\boldsymbol{I}| = |\boldsymbol{J}| = s$, shared across tasks at this layer, and define row / column selection matrices

$$\boldsymbol{S}_r \in \{0, 1\}^{s \times d}, \quad \boldsymbol{S}_c \in \{0, 1\}^{d \times s}. \tag{20}$$

By construction, $\boldsymbol{S}_r$ and $\boldsymbol{S}_c$ are *selection* matrices: each row of $\boldsymbol{S}_r$ (and each column of $\boldsymbol{S}_c$) contains exactly one entry equal to 1 and zeros elsewhere.

For theoretical consistency, we use the standard rescaled sampling operators

$$\boldsymbol{P}_r \triangleq \sqrt{\frac{d}{s}} \, \boldsymbol{S}_r \in \mathbb{R}^{s \times d}, \qquad \boldsymbol{P}_c \triangleq \sqrt{\frac{d}{s}} \, \boldsymbol{S}_c \in \mathbb{R}^{d \times s}, \tag{21}$$

so that $\mathbb{E}[\boldsymbol{P}_r^\top \boldsymbol{P}_r] = \boldsymbol{I}_d$ and $\mathbb{E}[P_c P_c^\top] = I_d$ under uniform sampling. We then form the *rescaled* sketched LoRA factors

*Table 1.* $\mathcal{O}(\cdot)$ time complexities. The cheapest method is highlighted in **bold** ($T, r \ll n$). For our method, $s$ denotes the sketch size. See Appendix B for the detailed derivations.

| Space | TA | Iso-C | TSV |
|---|---|---|---|
| Full | $n^2 T r$ | $n^3$ | $n^3 T$ |
| KnOTS | $n^3 T^2$ | $n^3 T^2 + n^2 T r$ | $n^3 T^2 + T^3 r^2 n$ |
| Core | $n^2 T r$ | $n^2 T r + T^3 r^3$ | $n^2 T r + T^4 r^3$ |
| **CircuMerge (Ours)** | $\boldsymbol{nTr \log r}$ | $\boldsymbol{nTr \log r}$ | $\boldsymbol{nTr \log r + s^2 Tr}$ |

for each task:

$$\bar{\boldsymbol{S}}_{\boldsymbol{A}}^{(t)} = \boldsymbol{P}_r \tilde{\boldsymbol{A}}^{(t)} \in \mathbb{R}^{s \times r}, \quad \bar{\boldsymbol{S}}_{\boldsymbol{B}}^{(t)} = \tilde{\boldsymbol{B}}^{(t)} \boldsymbol{P}_c \in \mathbb{R}^{r \times s}, \tag{22}$$

which induce the rescaled sketched update

$$\begin{aligned} \Delta \hat{\boldsymbol{W}}^{(t)} &= \boldsymbol{P}_r \Delta \boldsymbol{W}^{(t)} \boldsymbol{P}_c \\ &= \boldsymbol{P}_r \tilde{\boldsymbol{A}}^{(t)} \tilde{\boldsymbol{B}}^{(t)} \boldsymbol{P}_c \\ &= \bar{\boldsymbol{S}}_{\boldsymbol{A}}^{(t)} \bar{\boldsymbol{S}}_{\boldsymbol{B}}^{(t)} \in \mathbb{R}^{s \times s}. \end{aligned} \tag{23}$$

Instead of operating on full task vectors in $\mathbb{R}^{d^2}$, we fuse tasks in the sketch space by stacking vectorized sketches:

$$\boldsymbol{A} = \left[ \mathrm{vec}(\boldsymbol{S}_{\boldsymbol{A}}^{(1)}), \dots, \mathrm{vec}(\boldsymbol{S}_{\boldsymbol{A}}^{(T)}) \right] \in \mathbb{R}^{sr \times T}, \tag{24}$$

$$\boldsymbol{B} = \left[ \mathrm{vec}(\boldsymbol{S}_{\boldsymbol{B}}^{(1)}), \dots, \mathrm{vec}(\boldsymbol{S}_{\boldsymbol{B}}^{(T)}) \right] \in \mathbb{R}^{rs \times T}. \tag{25}$$

**Merging.** We apply any core-space / conflict-aware merging operator to $\mathcal{A}$ and $\mathcal{B}$ (e.g., TSV (Gargiulo et al., 2025)/TIES (Yadav et al., 2023)/DARE (Yu et al., 2024) methods) to obtain merged vectors $\tilde{a}^{\star} \in \mathbb{R}^{sr}$ and $\tilde{b}^{\star} \in \mathbb{R}^{rs}$, then reshape them and undo the global scaling after merging as

$$\boldsymbol{S}_{\boldsymbol{A}}^{\star} \triangleq \sqrt{\frac{s}{d}} \, \mathrm{reshape}(\tilde{\boldsymbol{a}}^{\star}) \in \mathbb{R}^{s \times r}, \tag{26}$$

$$\boldsymbol{S}_{\boldsymbol{B}}^{\star} \triangleq \sqrt{\frac{s}{d}} \, \mathrm{reshape}(\tilde{\boldsymbol{b}}^{\star}) \in \mathbb{R}^{r \times s}. \tag{27}$$

**Reconstruction.** To reconstruct full merged factors in the transformed space, we overwrite the sampled rows / columns using $(S_{A}^{\star}, S_{B}^{\star})$ and average the remaining entries:

$$(\tilde{\boldsymbol{A}}^{\star})_{i,:} = \begin{cases} (\boldsymbol{S}_{\boldsymbol{A}}^{\star})_{p,:}, & i = I_p, \ p \in \{1, \dots, s\}, \\ \frac{1}{T} \sum_{t=1}^{T} (\tilde{\boldsymbol{A}}^{(t)})_{i,:}, & i \notin I, \end{cases} \tag{28}$$

$$(\tilde{\boldsymbol{B}}^{\star})_{:,j} = \begin{cases} (\boldsymbol{S}_{\boldsymbol{B}}^{\star})_{:,q}, & j = J_q, \ q \in \{1, \dots, s\}, \\ \frac{1}{T} \sum_{t=1}^{T} (\tilde{\boldsymbol{B}}^{(t)})_{:,j}, & j \notin J. \end{cases} \tag{29}$$

**Recovery.** Finally, we map back to the standard LoRA parameterization:

$$\boldsymbol{A}^{\star} = \tilde{\boldsymbol{A}}^{\star} \boldsymbol{C}^{-1}, \quad \boldsymbol{B}^{\star} = \boldsymbol{C} \tilde{\boldsymbol{B}}^{\star}, \tag{30}$$

so that

$$\Delta \boldsymbol{W}^{\star} = \boldsymbol{A}^{\star} \boldsymbol{B}^{\star} = \left( \tilde{\boldsymbol{A}}^{\star} \boldsymbol{C}^{-1} \right) \left( \boldsymbol{C} \tilde{\boldsymbol{B}}^{\star} \right). \tag{31}$$

The resulting $(\boldsymbol{A}^{\star}, \boldsymbol{B}^{\star})$ can be loaded as a standard LoRA module without modifying inference code.

### 5.3. Computational Complexity Analysis

Table 1 reports the asymptotic merge-time complexities under the standard regime $T, r \ll n$, where $n$ denotes the layer width and $r$ the LoRA rank (and $s$ is the sketch size for our method). In the full parameter space, Iso-C (Marczak et al., 2025) and TSV (Gargiulo et al., 2025) incur cubic-time costs in $n$ (e.g., $\mathcal{O}(n^3)$ and $\mathcal{O}(n^3 T)$), while KnOTS (Stoica et al., 2025) further amplifies the overhead due to repeated large-scale subspace operations (e.g., $\mathcal{O}(n^3 T^2)$ terms). Core-space style (Panariello et al., 2025b) methods reduce the dependence on $n$ for TA (Ilharco et al., 2023) / TSV (Gargiulo et al., 2025), but still introduce high-order costs in $(T, r)$ from constructing and operating on shared bases (e.g., $\mathcal{O}(T^3 r^3)$ or $\mathcal{O}(T^4 r^3)$).

Our efficiency gain stems from two complementary factors. First, the high-complexity merging rules (e.g., Iso-C / TSV) are applied only to compact sketches of size $\{s, r\}$ or $\{r, s\}$, so expensive linear-algebraic primitives scale with $s$ rather than $n$; for TSV this yields an additional $\mathcal{O}(s^2 T r)$ term on top of sketch construction. Second, compared with common-basis decomposition methods, our randomized characterization avoids decompositions of large matrices altogether: the shared circulant transform enables fast structured multiplications (FFT-style), and the overall cost of producing the task-aligned representations scales as $\mathcal{O}(nTr \log r)$. As a result, CircuMerge achieves substantially lower merge-time complexity than prior approaches while preserving compatibility with existing merging rules.

## 6. Experimental Results

### 6.1. Experimental Setup

Unless otherwise specified, we follow the experimental protocol of Core Space Merging (Panariello et al., 2025b) (and its inherited KnOTS (Stoica et al., 2025) setup) for datasets, released checkpoints, evaluation metrics, and baseline hyperparameter tuning, ensuring a fair comparison under matched conditions. Additional experimental details are deferred to Appendix Section A.

*Table 2.* Normalized accuracies of merged models on NLI tasks for Llama 3 8B.

| Method | Space | SNLI | MNLI | SICK | QNLI | RTE | SCITAIL | Avg ($\Delta$Acc) | Time [s] |
|---|---|---|---|---|---|---|---|---|---|
| Abs. Acc. | | 92.50 | 90.31 | 91.58 | 94.49 | 89.86 | 96.52 | – | – |
| TA | Full | 93.57 | 95.28 | 87.96 | 68.71 | 100.0 | 96.73 | 90.38 (+0.00) | 9.7 |
| TIES | Full | **95.17** | **96.19** | 84.18 | 74.18 | **100.0** | 96.78 | 91.08 (+0.00) | 83 |
| | KnOTS | 91.82 | 94.19 | 92.97 | 78.57 | **100.0** | 97.61 | 92.53 (+1.45) | 2963 |
| | Core | 92.07 | 93.51 | **93.63** | 83.72 | 99.19 | 97.66 | 93.30 (+2.22) | 8.3 |
| | **CircuMerge (Ours)** | 92.49 | 92.83 | 93.15 | **84.32** | 99.70 | **97.73** | **93.37** (+2.29) | **3.7** |
| DARE-TIES | Full | **94.76** | **96.80** | 78.39 | 72.08 | 98.39 | 96.20 | 89.44 (+0.00) | 110 |
| | KnOTS | 91.62 | 96.72 | 74.90 | **84.75** | **99.48** | **99.13** | 91.10 (+1.66) | 3154 |
| | Core | 92.10 | 93.58 | 93.70 | 83.68 | 99.19 | 97.66 | 93.32 (+3.88) | 10.3 |
| | **CircuMerge (Ours)** | 91.76 | 94.17 | **94.13** | 84.26 | 99.27 | 98.25 | **93.64** (+4.20) | **4.7** |
| TSV | Full | 95.38 | 95.12 | 88.83 | 76.80 | 101.61 | 97.56 | 92.55 (+0.00) | 3372 |
| | KnOTS | 92.53 | 95.83 | 82.77 | 77.01 | 100.0 | 97.08 | 90.87 (-1.68) | 4630 |
| | Core | **95.86** | 95.70 | 89.25 | 83.89 | **102.42** | **97.86** | **94.16** (+1.61) | 14.3 |
| | **CircuMerge (Ours)** | 95.64 | **95.93** | **89.40** | **86.40** | 99.73 | 97.65 | 94.13 (+1.58) | **4.1** |
| Iso-C | Full | 55.00 | 39.04 | 76.54 | 55.90 | 46.77 | 69.25 | 57.08 (+0.00) | 543 |
| | KnOTS | 85.28 | 52.86 | 89.43 | 54.90 | 75.00 | 77.73 | 72.53 (+15.45) | 4844 |
| | Core | **91.54** | **90.10** | 87.87 | **75.85** | **99.19** | **97.42** | **90.33** (+33.25) | 11.6 |
| | **CircuMerge (Ours)** | 90.84 | 87.42 | **89.47** | 73.46 | 98.61 | 96.43 | 89.37 (+32.29) | **5.1** |

## 6.2. Main Results

**Language suite.** In the 6-task NLI benchmark with Llama 3 8B adapters, as reported in Table 2, CircuMerge achieves competitive performance in terms of normalized accuracy. It outperforms existing approaches in most configurations and obtains state-of-the-art results in several settings. CircuMerge consistently outperforms existing approaches in most configurations, achieving SOTA results in several settings. Specifically, across all tested spaces, CircuMerge not only matches the best performing methods in terms of accuracy but also surpasses them in certain cases, such as achieving a significant improvement in performance under the DARE-TIES configurations. Our CircuMerge method achieves SOTA-level accuracy across all settings while reducing the time consumption to approximately half of that required by traditional methods.

**Vision suite.** As shown in Table 3, on the 8-task CLIP vision benchmark, CircuMerge achieves competitive or superior average performance across different merging spaces, matching or outperforming the strongest baselines in most settings. The gains are particularly noticeable in configurations where subspace-based rules, such as TSV and Iso-C variants, are sensitive to task interference and subspace alignment. Importantly, sketching does not qualitatively change the relative behavior of different merge rules: methods that perform well in the original space typically remain strong after CircuMerge, whereas weaker methods still tend to underperform. This indicates that CircuMerge provides a low-cost and reliable sketch-space alternative that preserves the main empirical conclusions of the original merging rules.

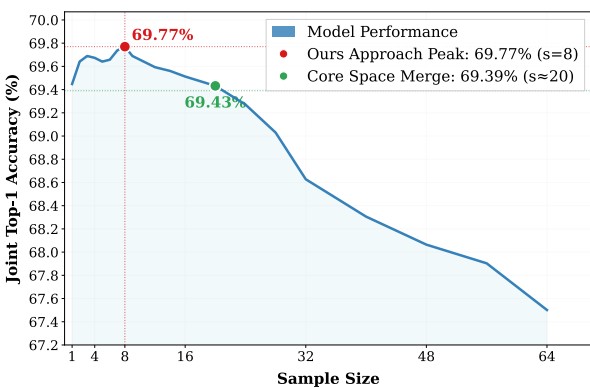

*Figure 2.* **Ablation on sketch size** $s$. We report average normalized accuracy (and optionally absolute accuracy) versus $s$.

**Efficiency: merge time and evaluation time.** Across language suites, our method significantly reduces merge time compared to operating on full adapters. All timing numbers are measured on a single RTX 3090 under identical software settings, ensuring that the reported speedups reflect algorithmic savings rather than differences in parallelism. Empirically, our merging procedure costs at most **44%** of the runtime of the corresponding baseline that operates on full adapters, indicating a consistent reduction in wall-clock overhead across tasks and suites.

## 6.3. Analysis and Ablations

**Effect of sketch size** $s$**.** We study how the sketch size $s$ (i.e., the number of samples used to form the sketches) trades off accuracy and efficiency. As shown in Figure 2,

*Table 3.* Normalized accuracies of merged models on the vision tasks with ViT-B/32.

| Method | Space | Cars | DTD | EuroSAT | GTSRB | MNIST | RESISC | SUN397 | SVHN | Avg (ΔAcc) |
|---|---|---|---|---|---|---|---|---|---|---|
| Abs. Acc. | | 74.00 | 58.30 | 99.00 | 92.70 | 99.30 | 88.40 | 64.50 | 96.20 | – |
| TA | Full | 81.97 | 73.72 | 48.97 | 42.24 | 53.12 | 71.50 | 97.46 | 41.25 | 63.78 (+0.00) |
| TIES | Full | 82.37 | 72.72 | 49.91 | 36.62 | 57.16 | 69.38 | **96.92** | 44.56 | 63.70 (+0.00) |
| | KnOTS | 83.75 | 74.45 | 50.36 | 47.31 | 67.01 | 71.79 | 96.51 | **50.64** | 67.73 (+4.03) |
| | Core | 84.74 | **76.46** | **52.19** | 50.41 | 67.36 | 71.21 | 96.45 | 50.18 | **68.63** (+4.93) |
| | **CircuMerge (Ours)** | **85.46** | 74.57 | 51.32 | **50.97** | **67.68** | **72.30** | 96.35 | 48.43 | 68.39 (+4.69) |
| DARE-TIES | Full | 82.14 | 73.72 | 49.35 | 37.78 | 56.63 | 70.14 | 97.35 | 42.12 | 63.65 (+0.00) |
| | KnOTS | 82.01 | 72.90 | 44.15 | 45.54 | 60.59 | 70.89 | 95.56 | 47.64 | 64.91 (+1.26) |
| | Core | 84.57 | **76.09** | **57.09** | 51.01 | 66.64 | 71.39 | **96.16** | 52.14 | 69.39 (+5.74) |
| | **CircuMerge (Ours)** | **85.67** | 72.87 | 53.46 | **53.91** | **70.68** | **72.64** | 95.43 | **53.49** | **69.77** (+6.12) |
| TSV | Full | 83.44 | **75.55** | 50.99 | 45.03 | **59.31** | 73.33 | 96.40 | 49.23 | 66.66 (+0.00) |
| | KnOTS | 81.86 | 74.91 | 51.25 | 41.64 | 53.93 | 71.64 | **97.95** | 40.36 | 64.19 (-2.47) |
| | Core | 83.86 | 75.09 | 52.64 | **45.39** | 58.53 | 72.95 | 97.63 | 45.21 | 66.41 (-0.25) |
| | **CircuMerge (Ours)** | **84.32** | 74.94 | **53.49** | 44.67 | 57.64 | **76.49** | 97.15 | **46.44** | **66.89** (+0.23) |
| CART | Full | 83.04 | 81.93 | 50.39 | 70.17 | 59.14 | 79.11 | **99.26** | 49.11 | 71.52 (+0.00) |
| | KnOTS | **83.94** | 75.18 | 52.23 | 54.48 | 64.78 | 74.48 | 95.88 | **55.73** | 69.59 (-1.93) |
| | Core | 80.83 | **83.94** | 54.99 | **73.28** | **66.25** | 80.95 | 98.69 | 48.57 | **73.44** (+1.92) |
| | **CircuMerge (Ours)** | 80.61 | 82.51 | **55.34** | 72.73 | 65.83 | **81.04** | 97.58 | 50.91 | 73.32 (+1.80) |
| TIES +Iso-C | Full | 78.86 | 74.45 | 60.01 | 39.02 | 66.65 | 70.30 | 98.39 | 48.59 | 67.03 (+0.00) |
| | KnOTS | 78.46 | 80.38 | **58.81** | 64.97 | **72.10** | 76.89 | 98.33 | 49.78 | 72.47 (+5.44) |
| | Core | **82.91** | 84.76 | 52.41 | 78.79 | 71.56 | 81.43 | 99.48 | 52.14 | 75.44 (+8.41) |
| | **CircuMerge (Ours)** | 81.83 | **85.34** | 48.86 | **84.37** | 69.82 | **83.49** | **99.74** | **52.61** | **75.76** (+8.73) |
| DARE-TIES +Iso-C | Full | 78.71 | 75.54 | 50.84 | 42.86 | 65.03 | 71.88 | 98.92 | 48.08 | 66.48 (+0.00) |
| | KnOTS | 82.93 | 74.18 | 49.31 | 46.73 | 66.64 | 71.82 | 96.72 | 50.57 | 67.36 (+0.88) |
| | Core | 83.27 | **83.12** | **54.55** | 79.04 | 71.83 | 82.08 | **99.36** | 52.37 | **75.70** (+9.22) |
| | **CircuMerge (Ours)** | **83.34** | 82.67 | 53.96 | 68.49 | **73.49** | **83.97** | 99.18 | **53.79** | 74.86 (+8.38) |
| TSV +Iso-C | Full | 79.38 | 80.38 | 57.99 | 65.64 | 64.22 | 79.74 | 98.59 | 46.49 | 71.55 (+0.00) |
| | KnOTS | 80.81 | 83.03 | **58.25** | 74.34 | 67.66 | 79.69 | 98.54 | 49.86 | 74.02 (+2.47) |
| | Core | **82.98** | 85.12 | 50.95 | 84.25 | 71.14 | 84.39 | 99.06 | 53.53 | 76.43 (+4.88) |
| | **CircuMerge (Ours)** | 82.46 | **85.79** | 57.64 | **86.09** | 69.16 | **84.92** | **99.28** | **54.93** | **77.53** (+5.98) |
| CART +Iso-C | Full | 80.33 | 82.11 | **57.31** | 77.38 | 71.17 | 81.57 | 98.72 | 51.91 | 75.06 (+0.00) |
| | KnOTS | 82.05 | 80.47 | 56.12 | 64.58 | 62.40 | 78.81 | 99.22 | 45.05 | 71.09 (-3.97) |
| | Core | 82.93 | 84.21 | 51.14 | **81.32** | 72.12 | 82.83 | **99.33** | 55.32 | 76.15 (+1.09) |
| | **CircuMerge (Ours)** | **83.07** | **84.61** | 49.43 | 81.18 | **72.69** | **83.79** | 99.28 | **56.94** | **76.37** (+1.31) |
| Iso-C | Full | 80.16 | 83.03 | 51.44 | 74.76 | 70.72 | 79.89 | 98.66 | 50.20 | 73.60 (+0.00) |
| | KnOTS | 80.33 | 79.29 | **57.50** | 67.60 | 65.63 | 79.54 | **99.26** | 46.62 | 71.97 (-1.63) |
| | Core | **83.35** | **84.30** | 50.13 | **81.97** | **71.07** | 83.46 | 99.17 | **53.90** | 75.92 (+2.32) |
| | **CircuMerge (Ours)** | 82.76 | 82.46 | 56.72 | 81.26 | 70.42 | **84.31** | 99.24 | 53.42 | **76.32** (+2.72) |

*Table 4.* Effect of the transform structure and sharing strategy. We report normalized/absolute accuracy and merge time under identical settings.

| Transform | Norm. Acc. (↑) | Merge Time (s) (↓) |
|---|---|---|
| None (direct sketching) | 57.92 | **3.46** |
| Random (invertible, shared) | 62.49 | 5.48 |
| Circulant (shared, ours) | **69.77** | 3.72 |

performance follows a clear non-monotonic trend: small sketches under-represent task updates, whereas overly large sketches introduce lower-energy or less-aligned components and weaken the regularizing effect of sketching. This behavior is consistent with the mechanism of CircuMerge. After SCT, dominant task-relevant directions are better aligned in the shared coordinate system, so a compact sketch is sufficient to capture the principal components of the merged updates. Increasing $s$ further improves coverage but may also incorporate weaker task-irrelevant directions, leading to additional noise. Thus, $s$ should balance coverage and

regularization rather than induce monotonic improvement. Overall, $s = 8$ yields the best average normalized accuracy (and strongest absolute accuracy) across both suites, so we use $s = 8$ as the default in all main experiments.

**Transform structure and sharing in rank space.** Table 4 compares different transform structures and sharing strategies in terms of normalized accuracy and merge time. The "None" variant directly performs sketching without rank-space transformation, achieving 57.92 normalized accuracy with 3.46 seconds of merge time. A random invertible shared transform improves accuracy to 62.49, showing the benefit of invertible rank-space mixing, but increases merge time to 5.48 seconds due to dense $O(r^2)$ multiplication. In contrast, our Circulant (shared) transform achieves the highest normalized accuracy of 69.77 while keeping the merge time at 3.72 seconds. This efficiency comes from the circulant structure, which supports FFT-based multiplication with $O(r \log r)$ complexity in the LoRA rank space. Since the transform is applied across all tasks and layers, this com-

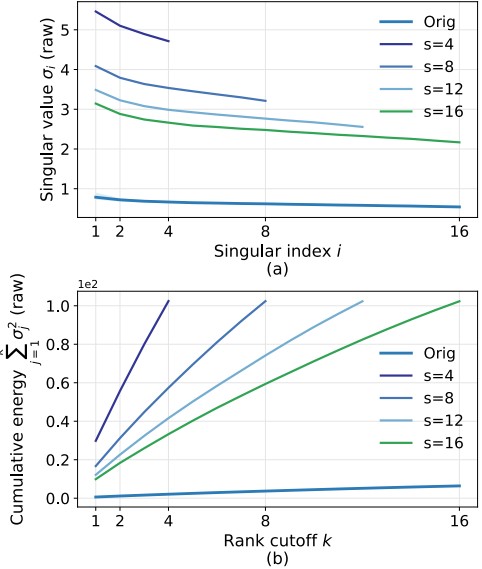

*Figure 3.* **Singular-spectrum dynamics under circulant reparameterization and rank sampling.** (a) Raw singular values $\sigma_i$ of the LoRA factor $A$ before sampling (Orig) and after SCT-enhanced sampling with different ranks $s \in \{4, 8, 12, 16\}$. (b) Corresponding cumulative energy $\sum_{j=1}^{k} \sigma_j^2$ as a function of rank cutoff $k$, demonstrating how SCT re-couples spectral components to achieve a near-linear energy distribution.

plexity gap accumulates, making SCT substantially faster than a general random transform while preserving effective rank-space alignment.

**Spectral dynamics.** To understand why our sketch-based design is effective, we directly probe how the shared circulant transform (SCT) reshapes the singular-value energy distribution of the matrix being fused. Concretely, we compare the original full matrix against SCT-based sampled reconstructions under varying sampling ranks $s$, and report (i) the singular-value profiles and (ii) the cumulative spectral energy curves. As shown in Figure 3(a), the singular values captured under different ranks $s$ remain markedly higher than those of the original full-rank matrix (Orig), indicating that energy typically lost by truncation is concentrated in the sampled subspace. Figure 3(b) further shows near-linear cumulative energy growth, confirming that SCT re-couples features across the spectrum and aggregates global energy. Crucially, random sampling after SCT does not simply "truncate" the singular-value spectrum; instead, it aggregates it.

## 7. Conclusion

This paper proposes **CircuMerge**, a sketch-based framework for efficient multi-task LoRA merging. CircuMerge applies a shared circulant transform in the LoRA rank space and performs merging on compact row-column sketches,

avoiding dense weight-delta merging and costly subspace factorizations. Using the Task Arithmetic objective as a tractable case, we establish a theoretical connection between merging quality and the effective rank of the matrices being merged, which motivates structured rank-space mixing before sketching. Extensive experiments on vision and language benchmarks show that CircuMerge reduces the overall merging time by at least 44% compared to state-of-the-art approaches while achieving optimal or near-optimal accuracy. Although our theoretical analysis is scoped to Task Arithmetic, the empirical results further show that CircuMerge consistently extends to different merging rules in practice. These results suggest that sketch-space merging with shared circulant transforms is an effective and scalable direction for integrating multiple task-specific LoRA adapters. Future work will explore more principled sketching strategies, extend the framework to broader PEFT modules, and evaluate its behavior on larger task collections.

## Acknowledgments

This work is supported in part by the Hebei Provincial Department of Science and Technology Foundation under grant (No.26240601D), the National Natural Science Foundation of China (No.62106259), the National Natural Science Foundation of China Joint Fund for Enterprise Innovation and Development (No.KZ77121701), the Local Science and Technology Development Fund of Hebei Province Guided by the Central Government of China through grant 254Z9902G, and Hebei Natural Science Foundation under grant F2024210008.

## Impact Statement

This paper presents work whose goal is to advance the field of Machine Learning. There are many potential societal consequences of our work, none which we feel must be specifically highlighted here.

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

# A. Additional Experimental Setup Details

**Benchmarks and checkpoints.** We evaluate on the same multi-task suites and the corresponding LoRA (Hu et al., 2022) checkpoints released by KnOTS (Stoica et al., 2025). For vision, we use CLIP (Radford et al., 2021) with ViT-B/32 and ViT-L/14 (Dosovitskiy et al., 2021) encoders and the standard 8-task suite: Cars196 (Krause et al., 2013), DTD (Cimpoi et al., 2014), EuroSAT (Helber et al., 2019), GTSRB (Stallkamp et al., 2011), MNIST (LeCun et al., 1998), RESISC45 (Cheng et al., 2017), SUN397 (Xiao et al., 2010), and SVHN (Netzer et al., 2011). For language, we use Llama 3 8B (Team, 2024) on the 6 NLI tasks: SNLI (Bowman et al., 2015), MNLI (Williams et al., 2018), SICK (Marelli et al., 2014), QNLI (Wang et al., 2019; Rajpurkar et al., 2016), RTE (Dagan et al., 2006), and SciTail (Khot et al., 2018).

**Baselines.** We compare against representative merging methods spanning weight-space and subspace-based approaches, including TA (Ilharco et al., 2023), TIES (Yadav et al., 2023), DARE-TIES (Yu et al., 2024), TSV (Gargiulo et al., 2025), CART (Choi et al., 2025), and Iso-C (Marczak et al., 2025). For methods that can be instantiated in different parameter spaces, we evaluate their standard variants in the Full update space, the KnOTS space (Stoica et al., 2025), and (iii) the Core Space (Panariello et al., 2025b).

**Evaluation metrics.** We report (i) *normalized accuracy*, defined as the ratio between the merged model accuracy and the corresponding single-task fine-tuned model accuracy for each task, and (ii) *absolute accuracy* on the merged model. To quantify efficiency, we additionally report *merge time*, measured as the wall-clock time to produce the merged adapter/model from the set of task-specific LoRA updates (excluding any one-time checkpoint loading when stated). All timing results are obtained on a single RTX 3090 under identical software settings.

**LoRA configuration.** LoRA rank is fixed to $r = 16$ and applied to all attention projections (Q/K/V/O) in every attention layer.

**Hyperparameter selection protocol.** All methods are tuned using the same validation-holdout linear search. The scaling factor $\alpha$ is searched from $0.1$ with step $0.1$. For TIES / DARE-TIES, Top-$K$ starts at 10 with step 10. For DARE-TIES, the pruning factor $p$ starts at $0.1$ with step $0.1$. For CART, the pruning rank is searched over $\{0.04, 0.08, 0.16, 0.32\}$. All tuned hyperparameters are selected under this shared protocol.

**Compute and timing protocol.** All experiments are run on a single NVIDIA RTX 3090 GPU (24GB). For fair runtime reporting, merge-time and evaluation latencies are measured on the same RTX 3090 under identical software settings.

**CircuMerge settings.** CircuMerge introduces a sketch size $s$ and a shared structured transform. Unless otherwise stated, $s$ is selected via the same validation protocol above, and one shared transform per layer is used across all tasks.

# B. Complexity Analysis of Sketch-Based LoRA Merging

**Notation.** We consider a single layer equipped with LoRA adapters for $T$ tasks. The dense layer weight is $W \in \mathbb{R}^{d_{\text{out}} \times d_{\text{in}}}$ and the $t$-th task adapter is

$$\Delta W_t = A_t B_t, \quad A_t \in \mathbb{R}^{d_{\text{out}} \times r}, \ B_t \in \mathbb{R}^{r \times d_{\text{in}}},$$

with LoRA rank $r \ll \min(d_{\text{out}}, d_{\text{in}})$. We use $d$ to denote a characteristic layer dimension, e.g., $d = \max(d_{\text{out}}, d_{\text{in}})$. The number of layers with LoRA is denoted by $L$. Our method introduces a shared circulant transform $C \in \mathbb{R}^{r \times r}$ on the rank dimension and constructs row / column sketches of size $s = \Theta(r)$ for each task.

We provide a step-by-step time complexity derivation for CircuMerge at a single LoRA-instrumented layer. Let $n$ be the layer width, $r$ the LoRA rank, $T$ the number of tasks, and $s$ the sketch size. We use standard RAM / BLAS accounting: dense matrix multiply of $(a \times b)$ by $(b \times c)$ costs $O(abc)$, and an FFT / IFFT of length $r$ costs $O(r \log r)$ up to constant factors.

**Step 1: Circulant reparameterization.** We fix an invertible circulant matrix $C \in \mathbb{R}^{r \times r}$ shared across tasks and compute $\tilde{A}^{(t)} = A^{(t)} C$ and $\tilde{B}^{(t)} = C^{-1} B^{(t)}$ for all $t \in \{1, \ldots, T\}$. Since $C$ is circulant, multiplying a length-$r$ vector by $C$ (or $C^{-1}$) can be implemented via FFTs in $O(r \log r)$ time.

For $\tilde{A}^{(t)} = A^{(t)}C$, we apply the length-$r$ circulant transform to each of the $n$ rows of $A^{(t)}$, so the cost per task is

$$\text{cost}\big(A^{(t)}C\big) = O(n \cdot r \log r) = O(nr \log r). \tag{32}$$

For $\tilde{B}^{(t)} = C^{-1}B^{(t)}$, we apply the transform to each of the $n$ columns of $B^{(t)}$, again costing $O(nr \log r)$ per task. Summing over all tasks gives

$$\text{cost(reparameterization)} = O(nrT \log r). \tag{33}$$

**Step 2: Row–column sketching.**  We select $s$ output indices $I$ and $s$ input indices $J$ (shared across tasks) and form $S_A^{(t)} = S_r \tilde{A}^{(t)} \in \mathbb{R}^{s \times r}$ and $S_B^{(t)} = \tilde{B}^{(t)} S_c \in \mathbb{R}^{r \times s}$. These are index-gather operations (copying $sr$ entries) rather than matrix multiplication, hence

$$\text{cost(sketch extraction per task)} = O(sr), \qquad \text{cost(sketch extraction)} = O(srT). \tag{34}$$

Note that explicitly forming $\Delta \hat{W}^{(t)} = S_A^{(t)} S_B^{(t)} \in \mathbb{R}^{s \times s}$ would cost $O(s^2 r)$ per task, but our pipeline operates on the factor sketches $(S_A^{(t)}, S_B^{(t)})$ and therefore does not require this product.

**Step 3: Fusion in sketch space.**  In CircuMerge, fusion is performed directly on the factor sketches $S_A^{(t)} \in \mathbb{R}^{s \times r}$ and $S_B^{(t)} \in \mathbb{R}^{r \times s}$. For notational uniformity, we apply the same fusion operator to $\{S_A^{(t)}\}_{t=1}^T$ and $\{(S_B^{(t)})^\top\}_{t=1}^T$, both of shape $s \times r$.

Following the full-space analysis of merging operators on an $m \times n$ matrix (Appendix B in (Panariello et al., 2025b)), and substituting $(m, n) = (s, r)$, the fusion-time overheads are:

$$\text{cost(fusion overhead)} = \begin{cases} O(srT), & \text{(TA: coordinate-wise arithmetic on } s \times r \text{ matrices),} \\ O(srT + s^2 r + r^3), & \text{(Iso-C: summation + SVD on an } s \times r \text{ matrix),} \\ O\big(T(s^2 r + r^3)\big), & \text{(TSV: SVD on } T \text{ matrices of size } s \times r\text{).} \end{cases} \tag{35}$$

We ignore constant factors from applying the operator to both $S_A^{(t)}$ and $(S_B^{(t)})^\top$.

**Step 4: Reconstruction (including the unsampled averages).**  We construct $\tilde{A}^\star \in \mathbb{R}^{n \times r}$ and $\tilde{B}^\star \in \mathbb{R}^{r \times n}$ by (i) overwriting the sampled rows / columns with merged sketches and (ii) averaging the remaining parts.

Overwriting costs $O(sr)$ and is negligible. The dominant reconstruction cost is the averaging of unsampled parts: for each $i \notin I$, computing $\frac{1}{T} \sum_{t=1}^T (\tilde{A}^{(t)})_{i,:}$ costs $O(rT)$, and there are $\Theta(n)$ such rows, giving $O(nrT)$. Similarly, averaging unsampled columns of $\tilde{B}$ costs another $O(nrT)$. Hence,

$$\text{cost(reconstruction)} = O(nrT). \tag{36}$$

**Step 5: Mapping back to standard LoRA parameters.**  Finally, we compute $A^\star = \tilde{A}^\star C^{-1}$ and $B^\star = C\tilde{B}^\star$. Each requires applying a length-$r$ circulant transform to $n$ rows / columns once, costing $O(nr \log r)$ per layer:

$$\text{cost(map back)} = O(nr \log r). \tag{37}$$

This term is not multiplied by $T$ and is lower-order whenever $T > 1$.

**Total time complexity.**  Combining (33), (34), (35), (36), and (37), the per-layer time complexity of CircuMerge is

$$\text{TA}: \quad O\big(nrT \log r + srT + nrT\big) = O\big(nrT \log r\big), \tag{38}$$

$$\text{Iso-C}: \quad O\big(nrT \log r + srT + nrT + (srT + s^2 r + r^3)\big) = O\big(nrT \log r\big), \tag{39}$$

$$\text{TSV}: \quad O\big(nrT \log r + srT + nrT + T(s^2 r + r^3)\big) = O\big(nrT \log r + s^2 rT\big), \tag{40}$$

where we drop the lower-order $O(nr \log r)$ map-back term in the last two expressions for readability.

## C. Expected SAR Gain from Spectral Uniformization via an Invertible Perturbation

### C.1. Setup and definitions

Fix one layer. Let task updates be $\{\Delta^{(t)}\}_{t=1}^{T} \subset \mathbb{R}^{d_{\text{out}} \times d_{\text{in}}}$ and define the task-arithmetic (TA) merge

$$M \triangleq \Delta^{\text{TA}} = \frac{1}{T} \sum_{t=1}^{T} \Delta^{(t)}. \tag{41}$$

Let $D \in \mathbb{R}^{d_{\text{out}} \times d_{\text{out}}}$ be an invertible (left) perturbation shared across tasks, and define the transformed TA merge

$$\widetilde{M} \triangleq \frac{1}{T} \sum_{t=1}^{T} D\Delta^{(t)} = D\Big(\frac{1}{T} \sum_{t=1}^{T} \Delta^{(t)}\Big) = DM. \tag{42}$$

**Principal subspace projector.** For any matrix $X$, let $U_X \in \mathbb{R}^{d_{\text{out}} \times d_{\text{out}}}$ denote the matrix of left singular vectors of $X$ (padding with an orthonormal completion if needed). Let $U_{X,1:k}$ be the first $k$ columns of $U_X$. Define the rank-$k$ projector onto the top-$k$ left singular subspace of $X$ by

$$\Pi_{k,X} \triangleq U_{X,1:k} U_{X,1:k}^{\top}. \tag{43}$$

$\varepsilon$-**energy rank.** Let $\{\sigma_i(X)\}_{i \geq 1}$ be singular values of $X$ in non-increasing order, and define the normalized Frobenius-energy spectrum

$$p_i(X) \triangleq \frac{\sigma_i^2(X)}{\|X\|_F^2}, \qquad \sum_{i \geq 1} p_i(X) = 1. \tag{44}$$

For a fixed $\varepsilon \in (0,1)$, define the $\varepsilon$-energy rank as

$$k_\varepsilon(X) \triangleq \min\Big\{k \geq 1 : \sum_{i \leq k} p_i(X) \geq 1 - \varepsilon\Big\}. \tag{45}$$

**SAR with adaptive rank.** Given a target update $\Delta$ and a reference matrix $X$, define

$$\text{SAR}_\varepsilon(\Delta, X) \triangleq \frac{\|\Pi_{k_\varepsilon(X),X}\Delta\|_F}{\|\Delta\|_F}. \tag{46}$$

This is the SAR induced by the top subspace of $X$ capturing $(1 - \varepsilon)$ of its Frobenius energy.

**Effective rank (spectral uniformity proxy).** We quantify "more uniform singular-value energy" using the participation ratio (Rényi-2 effective rank):

$$r_{\text{eff}}(X) \triangleq \frac{\big(\sum_i \sigma_i^2(X)\big)^2}{\sum_i \sigma_i^4(X)} = \frac{1}{\sum_i p_i^2(X)}. \tag{47}$$

Larger $r_{\text{eff}}(X)$ indicates a more spread-out (more uniform) energy distribution across singular directions.

### C.2. From spectral uniformity to larger $\varepsilon$-energy rank

**Lemma C.1** (Lower bound: $k_\varepsilon$ vs. effective rank). *For any matrix $X$ and any $\varepsilon \in (0,1)$,*

$$k_\varepsilon(X) \geq (1 - \varepsilon)^2 r_{\text{eff}}(X). \tag{48}$$

*Proof.* By Cauchy–Schwarz and (44),

$$\sum_{i \leq k} p_i(X) \leq \sqrt{k} \Big(\sum_{i \leq k} p_i^2(X)\Big)^{1/2} \leq \sqrt{k} \Big(\sum_{i \geq 1} p_i^2(X)\Big)^{1/2} = \sqrt{\frac{k}{r_{\text{eff}}(X)}}.$$

Let $k = k_\varepsilon(X)$. By (45), $\sum_{i \leq k_\varepsilon(X)} p_i(X) \geq 1 - \varepsilon$; hence

$$1 - \varepsilon \ \leq \ \sqrt{\frac{k_\varepsilon(X)}{r_{\text{eff}}(X)}} \quad \implies \quad k_\varepsilon(X) \ \geq \ (1-\varepsilon)^2 r_{\text{eff}}(X).$$

$\square$

### C.3. Isotropic projection: expected SAR depends only on subspace dimension

**Assumption C.2** (Left isotropy and independence). Let $\Delta \in \mathbb{R}^{d_{\text{out}} \times d_{\text{in}}}$ be a random matrix. Assume (i) *left isotropy*: $\mathbb{E}[\Delta\Delta^\top] = \alpha I_{d_{\text{out}}}$ for some $\alpha > 0$, and (ii) $\Delta$ is independent of the (random or deterministic) projector $\Pi$.

**Lemma C.3** (Expected projected energy under isotropy). *Under Assumption C.2, for any rank-k orthogonal projector $\Pi$,*

$$\mathbb{E}\big[\|\Pi\Delta\|_F^2\big] \ = \ \frac{k}{d_{out}} \, \mathbb{E}\big[\|\Delta\|_F^2\big]. \tag{49}$$

*Consequently, conditioning on any reference matrix $X$ (and its induced projector),*

$$\mathbb{E}\big[\text{SAR}_\varepsilon^2(\Delta, X) \mid X\big] \ = \ \frac{k_\varepsilon(X)}{d_{out}}. \tag{50}$$

*Proof.* Using $\|\Pi\Delta\|_F^2 = \text{tr}(\Delta^\top \Pi \Delta) = \text{tr}(\Pi\Delta\Delta^\top)$ and independence,

$$\mathbb{E}\big[\|\Pi\Delta\|_F^2\big] = \mathbb{E}\big[\text{tr}(\Pi\Delta\Delta^\top)\big] = \text{tr}\big(\Pi\,\mathbb{E}[\Delta\Delta^\top]\big) = \text{tr}(\Pi\,\alpha I) = \alpha\,\text{tr}(\Pi) = \alpha k.$$

Also $\mathbb{E}\|\Delta\|_F^2 = \mathbb{E}\text{tr}(\Delta\Delta^\top) = \text{tr}(\alpha I) = \alpha d_{\text{out}}$. Eliminating $\alpha$ yields (49). For (50), apply (49) to $\Pi = \Pi_{k_\varepsilon(X), X}$ and divide by $\|\Delta\|_F^2$ inside the expectation:

$$\mathbb{E}\big[\text{SAR}_\varepsilon^2(\Delta, X) \mid X\big] = \mathbb{E}\left[\frac{\|\Pi_{k_\varepsilon(X), X}\Delta\|_F^2}{\|\Delta\|_F^2} \ \middle| \ X\right] = \frac{k_\varepsilon(X)}{d_{\text{out}}}.$$

$\square$

### C.4. Main result: an invertible spectral "flattening" increases expected SAR after TA

**Theorem C.4** (Certified expected SAR gain from increased spectral uniformity). *Fix $\varepsilon \in (0, 1)$ and let $M = \Delta^{\text{TA}}$ be defined in (41). Let $D$ be invertible and $\widetilde{M} = DM$ as in (42). Assume Assumption C.2 holds for the target update $\Delta$ and that $\Delta$ is independent of the subspaces induced by $M$ and $\widetilde{M}$. Then:*

1. *(Exact expression)*

$$\mathbb{E}\Big[\text{SAR}_\varepsilon^2(\Delta, \widetilde{M})\Big] - \mathbb{E}\big[\text{SAR}_\varepsilon^2(\Delta, M)\big] \ = \ \frac{k_\varepsilon(\widetilde{M}) - k_\varepsilon(M)}{d_{out}}. \tag{51}$$

2. *(Certified lower bound in terms of effective rank)*

$$\mathbb{E}\big[\text{SAR}_\varepsilon^2(\Delta, X)\big] \ \geq \ \frac{(1-\varepsilon)^2}{d_{out}}\, r_{\text{eff}}(X), \qquad \forall X \in \{M, \widetilde{M}\}. \tag{52}$$

*In particular, if $r_{\text{eff}}(\widetilde{M}) > r_{\text{eff}}(M)$, then the certified lower bound on $\mathbb{E}[\text{SAR}_\varepsilon^2(\Delta, \widetilde{M})]$ is strictly larger than that on $\mathbb{E}[\text{SAR}_\varepsilon^2(\Delta, M)]$.*

*Moreover, if the spectral-uniformization is strong enough to ensure*

$$(1-\varepsilon)^2 r_{\text{eff}}(\widetilde{M}) \ > \ k_\varepsilon(M), \tag{53}$$

*then $k_\varepsilon(\widetilde{M}) > k_\varepsilon(M)$ and hence $\mathbb{E}[\text{SAR}_\varepsilon^2(\Delta, \widetilde{M})] > \mathbb{E}[\text{SAR}_\varepsilon^2(\Delta, M)]$.*

| Space | TIES | DARE-TIES | TSV | CART | TIES +Iso-C | DARE-TIES +Iso-C | TSV +Iso-C | CART +Iso-C | Iso-C |
|---|---|---|---|---|---|---|---|---|---|
| Full | 43.6 | 44.0 | 45.4 | 44.8 | 43.5 | 44.3 | 48.3 | 44.8 | 52.1 |
| KnOTS | 46.8 | 45.2 | 44.6 | 44.7 | 40.5 | 44.8 | 51.4 | 52.6 | 52.9 |
| Core | 47.4 | 47.6 | 44.5 | **49.6** | 54.1 | 54.0 | **55.7** | 55.6 | **55.9** |
| CircuMerge (Ours) | **47.6** | **48.3** | **46.1** | 48.7 | **56.3** | **55.8** | 54.4 | **56.7** | 53.4 |

*Table 5.* Joint-task setting absolute accuracy of merged models on the vision tasks with ViT-B/32.

*Proof.* (1) By Lemma C.3 with $X = \widetilde{M}$ and $X = M$,

$$\mathbb{E}\big[\text{SAR}_\varepsilon^2(\Delta, \widetilde{M})\big] = \frac{k_\varepsilon(\widetilde{M})}{d_{\text{out}}}, \qquad \mathbb{E}\big[\text{SAR}_\varepsilon^2(\Delta, M)\big] = \frac{k_\varepsilon(M)}{d_{\text{out}}},$$

which yields (51).

(2) Combine Lemma C.3 with Lemma C.1:

$$\mathbb{E}\big[\text{SAR}_\varepsilon^2(\Delta, X)\big] = \frac{k_\varepsilon(X)}{d_{\text{out}}} \geq \frac{(1-\varepsilon)^2}{d_{\text{out}}} r_{\text{eff}}(X).$$

This proves (52). Finally, if (53) holds, then by $k_\varepsilon(\widetilde{M}) \geq (1-\varepsilon)^2 r_{\text{eff}}(\widetilde{M})$ we obtain $k_\varepsilon(\widetilde{M}) > k_\varepsilon(M)$, and the strict expected SAR inequality follows from (1). □

**Interpretation.** Theorem C.4 formalizes the mechanism: an invertible shared perturbation $D$ that makes the TA spectrum more uniform (larger $r_{\text{eff}}(DM)$) increases the *certified* lower bound on expected SAR (and yields a strict improvement once the increase is large enough to raise the $\varepsilon$-energy rank $k_\varepsilon$). In practice, $r_{\text{eff}}(DM)$ and $k_\varepsilon(DM)$ are directly computable from singular values, so the condition can be verified empirically layer-wise.

### C.5. Joint-task evaluation with ViT-B/32.

We follow the joint-task vision protocol of prior work (Stoica et al., 2025; Panariello et al., 2025b), where the task identity is *unknown* at inference time. Concretely, the merged ViT-B/32 model is evaluated on the *union of all classes* across the considered vision datasets, so predictions are made over a single shared label space rather than per-task heads. This setting is substantially more challenging than standard multi-task evaluation, as the model must simultaneously separate categories from different tasks without access to a task ID. Table 5 reports the resulting absolute accuracy for different merging rules (TIES, DARE-TIES, TSV, CART) under several merging spaces (Full, KnOTS, Core, and CircuMerge), including their combinations with Iso-C.

Our results demonstrate that CircuMerge outperforms or matches existing methods across most merging configurations. Notably, CircuMerge achieves the highest accuracy when combined with Iso-C for multiple merging spaces, with a peak accuracy of **56.3%** on the Full space (compared to 52.1% with Iso-C alone). In comparison, traditional methods like TIES and DARE-TIES exhibit more modest improvements when combined with Iso-C, achieving a maximum of 48.3% and 44.3% accuracy, respectively. The **KnOTS** and **Core** spaces also benefit significantly from CircuMerge, with notable improvements in accuracy over baseline methods. In particular, CircuMerge achieves **48.3%** on DARE-TIES + Iso-C, compared to **47.6%** for Core and **46.1%** for TSV. Additionally, CircuMerge achieves a remarkable **56.7%** accuracy when applied to the Core space with Iso-C, indicating the method's robustness across various model architectures. In summary, CircuMerge consistently delivers superior performance, highlighting its efficiency and effectiveness for multi-task merging, particularly in settings where task identity is unknown and class overlap is significant.

