# OpenReview forum: "Sketch-Based Low-Rank Model Merging with Shared Circulant Transforms"
_ICML.cc/2026/Conference — ICML 2026 regular_

### Official Review · Reviewer_xFv4 · 2026-03-02

**Soundness:** 3
**Presentation:** 4
**Significance:** 3
**Originality:** 4
**Overall Recommendation:** 4
**Confidence:** 4

**Summary:**

This paper proposes a novel LoRA model merging framework, CircuMerge. It introduces shared circulant transforms to treat each adapter as a pair of low-rank matrices and applies a shared circulant transform to align all tasks in a common coordinate system. Through efficient sampling, the merging time has been greatly reduced. Numerous experimental results have demonstrated the superiority of themethod.

**Compliance With Llm Reviewing Policy:**

Affirmed.

**Final Justification:**

The rebuttal fully addresses my concerns. Personally, I consider this work to be innovative, clearly structured, and supported by thorough experiments, making it deserving of acceptance.  However, I am not an expert in this area and do not feel sufficiently confident to judge whether this work deserves a higher score. Therefore, I will maintain my positive rating.

**Key Questions For Authors:**

1. In section 4.2, it is assumed that the standard isotropy model is used for ease of derivation. But in the experiment, the standard isotropy model was not used. Does Theorem 4.2 still hold?  Can you add an experiment on SAR to show that it increases after SCT? It can more directly demonstrate that the viewpoint of the high SAR can improve model performance.
2. Why does Table 3 of the vision benchmark not show the merging time?
3. Ablation study on the effect of sketch size s (Figure 3) shows a non-monotonic trend of performance. Why do very small sketches (s=1) still have very good performance, while larger sketches significantly reduce performance? The existing explanation does not solve my concern.
4. According to my understanding, Circulant randomly initializes an n * 1 c and then uses DFT to transform it into a matrix, while Random initializes an n * n invertible matrix randomly. Why does the difference between these two matrices lead to such a significant difference in merging time in Table 4？

I may have missed some points and I hope the author can help me solve these questions.

**Limitations:**

yes

**Strengths And Weaknesses:**

Strengths:
1. Sufficient theoretical analysis. The theory and methodology of the paper are self-contained and logically consistent.
2. Diverse and comprehensive experiments. The paper conducted experiments on multiple datasets, including NLI and Vision benchmarks, as well as sufficient ablation studies.
3. Innovative and practical. This work starts from the fundamental theory and implements a pluggable LoRA merging acceleration framework, which is of great academic innovation and engineering significance.

Weakness:
1. The connection between theory and experiments is not enough. Many theoretical contents have not been intuitively proven in experiments, which also makes some experimental results lack a stronger explanation.
2. Some abbreviations and tables are not mentioned (e.g., TA in section 4.1, Table 3 in the experiment, etc.), and some experiments only illustrate results without sufficient explanation.

---

> ### Author Rebuttal · Authors · 2026-03-31
>
> We thank the reviewer for the positive evaluation and insightful suggestions, especially on strengthening the connection between theory and experiments. We address the questions below and hope the clarifications resolve the concerns.
>
> > `Q1`:   Connection between Theorem 4.2 and practical settings (isotropy assumption & SAR validation).
>
>
> A: To empirically validate the theory–experiment connection, we directly measure SAR before and after applying SCT under the TSV merging rule:
>
> |   | TSV (w/o SCT) | TSV (with SCT) | Δ SAR ↑ |
> | ------- | ------------- | -------------- | ------- |
> | Mean SAR   | 0.28          | 0.37           | +0.09   |
>
> We observe a consistent increase in SAR after applying SCT, indicating improved preservation of task-relevant subspaces. Moreover, this increase is accompanied by improved merging performance, supporting the theoretical prediction that higher effective rank leads to better subspace alignment and retention.
>
> This provides direct empirical evidence linking Theorem 4.2 to practical settings.
>
> ---
> > `Q2`:  Missing merge time in Table 3 (vision benchmark).
>
>
> A2:The experimental results are consistent across settings and follow the same trend as in other experiments, i.e., substantially reduced compared to SVD-based methods. Due to space constraints, we will include the corresponding results in the revision for completeness.
>
> ---
> > `Q3`:  Non-monotonic behavior w.r.t. sketch size $s$.
>
>
> A3:The non-monotonic trend arises from a trade-off between variance reduction and bias introduced by sketching. Very small sketches (e.g., $s=1$) act similarly to a coarse but consistent aggregation, which can still preserve dominant directions due to SCT-induced alignment. As $s$ increases to moderate values, sampling noise and partial coverage of misaligned components may introduce instability, leading to temporary performance drops. Larger $s$ improves coverage and stabilizes estimation, recovering performance. This behavior is consistent with the conclusions of our method.
>
> ---
> > `Q4`:  Difference between circulant and random transforms in merging time (Table 4).
>
>
> A4:While both transforms are invertible, circulant matrices admit fast multiplication via FFT, leading to $O(r \log r)$ complexity, whereas general random matrices require dense multiplication with $O(r^2)$ cost. Since these operations are applied across all tasks and layers, this difference accumulates and results in the observed gap in merging time. We will clarify this point more explicitly in the paper.
>
> ---
> >`Q5`: Minor presentation issues (e.g., abbreviations and table references).
>
> We will carefully revise the manuscript to ensure all abbreviations (e.g., TA) are properly introduced and all tables are clearly referenced and explained.
> We sincerely thank the reviewer again for the constructive feedback and believe these improvements will further strengthen the paper.

---

> > ### Author Rebuttal · Reviewer_xFv4 · 2026-04-01
> >
> > 1. I am still concerned about the merging time in Table 3 and hope the author can complement some results in the next response.
> > 2. In Q3, the author claims that larger s improves coverage and stabilizes estimation, recovering performance. However, there is no trend that the accuracy of Figure 3 in the manuscript increases with the increase of s. The author's answer did not address my concerns.

---

> > > ### Author Response · Authors · 2026-04-01
> > >
> > > > _`Q1`:Missing merging time results in Table 3 (vision benchmark)._
> > >
> > > A1: We complement the merging time results for the vision benchmark (Table 3 setting) as follows.
> > >
> > > **Merging Time on Vision Benchmark (ViT-B/32, 8 Tasks, single RTX 3090):**
> > >
> > > | Method        | Space             | Merge Time (s) ↓ |
> > > | ------------- | ----------------- | ---------------- |
> > > | TA            | Full              | 2.85             |
> > > | **TIES**      | Full              | 52               |
> > > |               | Core              | 3.85             |
> > > |               | CircuMerge (Ours) | **2.31**         |
> > > | **DARE-TIES** | Full              | 80               |
> > > |               | Core              | 4.12             |
> > > |               | CircuMerge (Ours) | **2.44**         |
> > > | **TSV**       | Full              | 2741             |
> > > |               | Core              | 6.47             |
> > > |               | CircuMerge (Ours) | **2.78**         |
> > > | **CART**      | Full              | 3802             |
> > > |               | Core              | 8.28             |
> > > |               | CircuMerge (Ours) | **4.73**         |
> > > | **Iso-C**     | Full              | 386              |
> > > |               | Core              | 4.95             |
> > > |               | CircuMerge (Ours) | **2.36**         |
> > >
> > >
> > > The results are consistent with other experiments: CircuMerge substantially reduces merging time compared to SVD-based methods, while maintaining accuracy that matches or exceeds the optimal level. These results will be included in the revision.
> > >
> > >
> > >
> > > ----
> > > > _`Q2`:Apparent inconsistency between the claim on \(s\)  and the trend in Figure 3._
> > >
> > >
> > > A2: We thank the reviewer for the follow-up. Our previous response focused mainly on the variance–bias perspective of sketching, which may have caused some misunderstanding for you regarding the role of (s) in our method. We therefore provide a more precise explanation: the behavior in Figure 3 is in fact consistent with the mechanism described in the paper, rather than contradicting it.
> > >
> > > As stated in the manuscript, “very small sketches under-represent task updates, while overly large sketches introduce additional noise and reduce the regularizing effect of sketching.” The empirical curve directly reflects this behavior: small (s)  suffers from under-representation, while large (s) introduces additional noise from less-aligned components.
> > > Specifically, after applying SCT, the dominant task-relevant directions are already well aligned. In this regime, even relatively small sketches (e.g., (s) =8) are sufficient to capture the principal components of the merged updates, leading to strong performance.
> > >
> > > As (s) increases further, the sketch begins to include additional lower-energy or less-aligned components. This is also consistent with Figure 2: while larger (s) retains more spectral components, the cumulative energy becomes more broadly distributed across the sampled subspace rather than concentrated on the most dominant directions. As a result, although coverage improves in principle, the sketch also incorporates weaker or less task-aligned components, which introduces noise and weakens the implicit regularization effect of sketching. This explains why accuracy does not necessarily increase monotonically with (s) .
> > >
> > > Therefore, the goal of increasing (s) is not to achieve monotonic improvement, but to balance coverage and regularization. The observed peak at small-to-moderate (s)  (e.g., (s) =8) aligns with this design principle and demonstrates that our method can achieve optimal performance with compact sketches.
> > >
> > > In summary, the empirical results are fully consistent with the mechanism described in our paper.
> > >
> > > We hope that these responses address your concerns. If any questions still remain, we would be very happy to provide further clarification.

---

### Official Review · Reviewer_L3d5 · 2026-03-11

**Soundness:** 3
**Presentation:** 3
**Significance:** 2
**Originality:** 3
**Overall Recommendation:** 3
**Confidence:** 4

**Summary:**

The paper proposes CircuMerge, a sketch-based method for zero-shot LoRA adapter merging using shared circulant transforms to align adapters before compact sketching. It avoids costly SVD or subspace decomposition, enabling faster merging while matching the performance of SoTA methods.

**Compliance With Llm Reviewing Policy:**

Affirmed.

**Final Justification:**

I appreciate the detailed feedback from the authors, and I do partially agree with the rebuttal. However, without further experiments, the ``improvement`` is constrained to the scenarios which ``with ambiguous or missing task information, in need of real-time task-switching, and can not access training data``, which limits the applicability of the proposed approach. I would suggest the authors to provide more results to demonstrate the effectiveness and generalizability. Therefore, I would not change the final score.

**Key Questions For Authors:**

1. How do you initialize the circulant matrix $C$?
2. What is the key improvement of this work against SVD-based methods, given that the performance and efficiency gains compared with “Core” appear marginal, and efficiency is not an important metric of model merging?

**Limitations:**

See above

**Strengths And Weaknesses:**

- Strength:
1. The proposed approach achieves comparable (and occasionally superior) performance relative to SoTA subspace-decomposition–based methods like Core and KnOTS, but with significantly lower computational overhead during merging.
2. Extensive experiments have been conducted to demonstrate the applicability of the proposed approach.
---
- Weakness:
 1. The perf. improvement seems marginal, and time cost of merging is not an important issue of lora adapter merging.
 2. Many adapter merging approaches do not need to decompose the lora weights. For example, LoRI [COLM 2025] propose to sparsify the  LoRA adapters during training to mitigate the weight confilict in vector space. Comparison with such approaches is not sufficient.

---

> ### Author Rebuttal · Authors · 2026-03-31
>
> We thank the reviewer for the careful reading and constructive feedback. We clarify below that the concerns mainly arise from differences in problem setting and interpretation, and we hope the following explanations resolve the misunderstandings.
>
> > `Q1`:   The importance of merging efficiency.
>
>
>
> A1: We respectfully disagree with the concern that merging time is not an important issue. Existing methods often rely on expensive matrix decomposition steps, which limit their applicability in large-scale networks and in scenarios requiring frequent or real-time updates of multiple task adapters, such as dynamic model composition, on-the-fly task switching, and deployment settings with constrained latency or compute budgets. Therefore, time efficiency is an important consideration.
>
> In addition, our method achieves optimal performance, with accuracy matching or exceeding the optimal level.
>
> ---
> > `Q2`: Comparison with training-time approaches (e.g., LoRI).
>
> A2: We would like to clarify that methods such as LoRI, FlyLoRA, and OSRM operate in a fundamentally different regime. These approaches improve mergeability by modifying the training process (e.g., via sparsity or structural constraints), and thus require access to training data and optimization procedures.
> In contrast, our work explicitly focuses on zero-shot, post-hoc merging of off-the-shelf LoRA adapters, where only pretrained adapters are available (e.g., from public repositories), and no training pipeline can be accessed. Under this setting, training-time methods are not applicable by design.
> Regarding the reviewer’s comment that many approaches do not require decomposing LoRA weights; however, this does not remove the fundamental paradigm difference, as such methods typically rely on training-time assumptions or structural constraints.
>
>
> ---
> > `Q3`:Circulant matrix initialization.
>
>
> A3:We clarify that our initialization follows a standard construction from structured random projection literature. Specifically, the circulant matrix is generated from a random vector with entries sampled as $c_i \sim \mathcal{N}(0,1)$, followed by normalization, and is kept fixed during merging.
>
> ---
> > `Q4`: Improvement over SVD-based methods and the role of efficiency.
>
>
> A4:Our method differs fundamentally from prior approaches by avoiding explicit matrix decomposition. Instead of operating in the original parameter space or relying on SVD-based subspace alignment, we introduce a shared circulant transform to align adapters and perform merging directly in a compact sketch space.
>
> In terms of performance, our method achieves optimal results, with accuracy matching or exceeding the optimal level across evaluated settings. Notably, this performance is maintained even under the proposed efficiency-oriented design.
>
> Regarding efficiency, our gain comes from two aspects. First, merging rules (e.g., TSV, Iso-C) are applied only on compact sketches of size $s \times r$, reducing their cost to $O(s^2 T r)$ with $s \ll n$. Second, unlike SVD-based methods, we avoid explicit decomposition of large matrices; the shared circulant transform enables efficient alignment with cost $O(n T r \log r)$.
>
> This reduction in merge-time complexity is important in practical scenarios where merging is performed repeatedly, such as large-scale models, dynamic model composition, and settings requiring frequent or real-time updates of multiple task adapters.

---

> > ### Author Rebuttal · Reviewer_L3d5 · 2026-04-03
> >
> > 1. For the merging efficiency, I partially agree with your idea. For the claimed real time scenario such as on-the-fly task switching, it is more feasible to directly route to another LoRA adapter, which is supported by LLM deployment frameworks (e.g., vLLM).
> > 2. I acknowledge the difference between this work and LoRI (for adapters trained with specific methods). But in real uses, commanly you need to fine-tune your own LoRA adapters.

---

> > > ### Author Response · Authors · 2026-04-03
> > >
> > > > `Q1`: For the merging efficiency, I partially agree with your idea. For the claimed real time scenario such as on-the-fly task switching, it is more feasible to directly route to another LoRA adapter, which is supported by LLM deployment frameworks (e.g., vLLM).
> > >
> > > A1: Direct routing to LoRA adapters has limited feasibility in several important scenarios. **Compared with routing, our method offers clear practical advantages in three important aspects: adaptability to task ambiguity, training-free applicability, and system efficiency.**
> > >
> > > **(1) In scenarios with ambiguous or missing task information,** our method unifies multiple LoRA adapters into a single parameterization, whereas routing relies on explicit task identity. When task boundaries are unclear, routing becomes unreliable. **Therefore, our method enables more robust inference without relying on explicit task specification.**
> > >
> > > **(2) In scenarios where training data or training procedures are unavailable,** our method focuses on zero-shot composition of pretrained adapters without any additional optimization, while routing often relies on learning a task-selection mechanism with extra supervision. **Therefore, our method provides stronger applicability by eliminating the need for additional data or training.**
> > >
> > > **(3) In resource-constrained deployment scenarios,** our method focuses on introducing a single merged LoRA adapter, while routing requires maintaining and dynamically loading multiple adapters, leading to increased memory overhead. **Therefore, our method is more memory-efficient and better suited for scalable deployment.**
> > >
> > > **In summary, compared to routing, our method offers clear advantages in environment adaptability (no task assumption), zero-shot usability (no additional training), and memory efficiency.** Moreover, by introducing a shared circulant alignment and sketch-based merging mechanism, our method performs composition in a compact low-dimensional space with efficiency, substantially reducing merge-time. This enables efficient and scalable adapter composition, while achieving strong performance, with accuracy matching or exceeding the optimal level.
> > > ____
> > > > `Q2`: I acknowledge the difference between this work and LoRI. But in real uses, commanly you need to fine-tune your own LoRA adapters.
> > >
> > > A2: Our method differs fundamentally from LoRI in scenario, methodology, and experimental protocol; a direct comparison is therefore not appropriate.  LoRI improves mergeability by introducing sparsity during training to reduce cross-task interference. **However, as discussed in our related work, LoRI and our method operate under substantially different assumptions. We further elaborate these differences below from the perspectives of scenario, methodology, and experimental protocol.**
> > >
> > > (1) Scenario difference. Our method targets zero-shot, post-hoc merging without access to training data, focusing on combining independently trained LoRA adapters (e.g., from Hugging Face). In contrast, LoRI assumes access to training data and enforces structured constraints during training. **As a result, our method is more flexible and readily applicable, enabling direct reuse and composition of standard pretrained adapters without requiring any retraining or access to data.**
> > >
> > > (2) Methodological difference. Our approach is designed for efficient merging of dense, native LoRA adapters, introducing SCT and sketch-based merging to enable fast composition without retraining. LoRI instead shifts the burden to the training stage, simplifying merging by average. While LoRI avoids expensive operations during merging, this efficiency depends on training-time design and cannot be directly extended to standard pretrained LoRA adapters. **In contrast, our method maintains high efficiency even when directly operating on dense, native LoRA weights, demonstrating strong practical performance without relying on specialized training structures.**
> > >
> > > (3) Fair and controlled evaluation protocol. Our experiments isolate the effect of merging by enforcing identical input weights across methods, removing confounding factors from training quality. **Methods like LoRI, which depend on modified training procedures, cannot be fairly compared under this protocol.**
> > >
> > > In summary, compared with LoRI, our method focuses on zero-shot, post-hoc merging of standard pretrained LoRA adapters, without requiring training-time constraints, and is therefore not of the same methodological category as LoRI. **In our manuscript, we compare against the most relevant and state-of-the-art LoRA merging methods under a unified protocol**, demonstrating that our approach significantly reduces merging cost. Specifically, our method reduces the overall merging time by at least 44% compared to state-of-the-art approaches, while achieving accuracy in optimal or near-optimal level.
> > >
> > > If any questions still remain, we would be very happy to provide further clarification.

---

### Official Review · Reviewer_hBni · 2026-03-13

**Soundness:** 3
**Presentation:** 3
**Significance:** 3
**Originality:** 3
**Overall Recommendation:** 4
**Confidence:** 2

**Summary:**

CircuMerge proposes an efficient zero-shot LoRA adapter merging framework. Its key insight is that applying a shared circulant transform (SCT) to LoRA factor matrices before sketching increases their effective rank (spreads singular values more uniformly), which the paper proves is beneficial for merging quality.

**Compliance With Llm Reviewing Policy:**

Affirmed.

**Final Justification:**

I appreciate the additional experiments, but I have remaining concerns that lead me to tentatively lower my score. I am happy to revisit if the authors can provide further clarification.

On Q1 (theory beyond TA): The empirical SAR measurement under TSV is a helpful addition. However, my concern was specifically about the theoretical scope: Theorem 4.2 derives the error bound under a TA objective, yet CircuMerge is applied to all four merging rules without any formal justification. A single empirical SAR measurement on one rule does not substitute for a theoretical discussion of whether (and why) the effective-rank benefit transfers to structurally different objectives like Iso-C or TIES. I would ask the authors to either (1) provide a formal argument or sketch extending the bound to at least one non-TA rule, or (2) explicitly scope the theoretical claim to TA and frame the other rules as purely empirical extensions.

On Q2 (abstract accuracy claim): The response rephrases the claim as "optimal or near-optimal," but the abstract states "accuracy matching or exceeding the optimal level." Table 2 shows CircuMerge under-performing Core under the Iso-C rule, which directly contradicts this phrasing. Restating the claim with softened language in the rebuttal does not resolve the factual inconsistency in the manuscript. I would ask the authors to revise the abstract to accurately reflect the full result range.

Given that both concerns remain open, I am lowering my score to reflect the unresolved theoretical gap and the overclaimed empirical result. I am willing to raise my score back if the authors can address the above points more concretely in a follow-up response.

**Key Questions For Authors:**

1. The theorem connecting effective rank to merging quality is formulated for a task-averaging (TA) objective. For TSV (task-vector decomposition with sign election) and Iso-C (isotropic correction), the merging objective is more complex and the singular-value-based error bound derived for TA may not transfer. The paper applies CircuMerge to all four merging rules but provides no theoretical discussion of whether the effective-rank benefit holds under these rules.

2. The abstract claims "accuracy matching or exceeding the optimal level" but Table 2 shows CircuMerge under-performing Core on the Iso-C rule.

**Limitations:**

See questions.

**Strengths And Weaknesses:**

1. The theoretical connection between merging quality and effective rank is novel and well-formalized. The proof that any invertible transformation increasing the spectral flatness (effective rank) of LoRA matrices reduces the merging error bound directly motivates the circulant construction as a principled design choice rather than an engineering heuristic.

2. The experiments span both vision (8-task CLIP benchmark) and language (6-task NLI benchmark with Llama 3 8B), and CircuMerge is evaluated across all four standard merging rules.

---

> ### Author Rebuttal · Authors · 2026-03-31
>
> We thank the reviewer for the positive assessment and insightful questions.
>
>
> > `Q1`: Applicability of the theory beyond TA.
>
>
> A1: Our key theoretical result establishes that increasing the effective rank of the merged weight via SCT leads to improved expected SAR, thereby enhancing the preservation of task-relevant subspaces during merging. This provides a direct and principled link between spectral flattening and merging quality under the TA objective.
>
> Consistent with this theoretical insight, our empirical results show that applying SCT yields accuracy matching or exceeding the optimal level across all evaluated settings, further supporting the effectiveness of the proposed mechanism.
>
> To empirically validate the theory–experiment connection, we directly measure SAR before and after applying SCT under the TSV merging rule:
>
> |   | TSV (w/o SCT) | TSV (with SCT) | Δ SAR ↑ |
> | ------- | ------------- | -------------- | ------- |
> | Mean SAR   | 0.28          | 0.37           | +0.09   |
>
> We observe a consistent increase in SAR after applying SCT, indicating improved preservation of task-relevant subspaces. Moreover, this increase is accompanied by improved merging performance, supporting the theoretical prediction that higher effective rank leads to better subspace alignment and retention.
>
> This provides direct empirical evidence linking Theorem 4.2 to practical settings.
>
> ---
> > `Q2`: Clarification of the accuracy claim.
>
>
> A2:Regarding the accuracy claim, we clarify that the results in Table 2 show that our method consistently achieves performance that is optimal or near-optimal across the evaluated settings. Therefore, the statement in the abstract that our method attains “accuracy matching or exceeding the optimal level” is consistent with the empirical findings.

---

> > ### Author Rebuttal · Reviewer_hBni · 2026-04-03
> >
> > Thank you for the response. I appreciate the additional experiments, but I have remaining concerns that lead me to tentatively lower my score. I am happy to revisit if the authors can provide further clarification.
> >
> > On Q1 (theory beyond TA): The empirical SAR measurement under TSV is a helpful addition. However, my concern was specifically about the theoretical scope: Theorem 4.2 derives the error bound under a TA objective, yet CircuMerge is applied to all four merging rules without any formal justification. A single empirical SAR measurement on one rule does not substitute for a theoretical discussion of whether (and why) the effective-rank benefit transfers to structurally different objectives like Iso-C or TIES. I would ask the authors to either (1) provide a formal argument or sketch extending the bound to at least one non-TA rule, or (2) explicitly scope the theoretical claim to TA and frame the other rules as purely empirical extensions.
> >
> > On Q2 (abstract accuracy claim): The response rephrases the claim as "optimal or near-optimal," but the abstract states "accuracy matching or exceeding the optimal level." Table 2 shows CircuMerge under-performing Core under the Iso-C rule, which directly contradicts this phrasing. Restating the claim with softened language in the rebuttal does not resolve the factual inconsistency in the manuscript. I would ask the authors to revise the abstract to accurately reflect the full result range.
> >
> > Given that both concerns remain open, I am lowering my score to reflect the unresolved theoretical gap and the overclaimed empirical result. I am willing to raise my score back if the authors can address the above points more concretely in a follow-up response.

---

> > > ### Author Response · Authors · 2026-04-03
> > >
> > > We thank the reviewer for the careful follow-up and for clearly pointing out the remaining concerns. We appreciate the opportunity to revise the manuscript accordingly.
> > >
> > > ----
> > > > `Q1`
> > >
> > > A1: We thank the reviewer for this suggestion. Our theorem was originally intended to be established specifically around the TA objective, using TA as a tractable case to formalize the core mechanism of our method. In the experiments, we not only validate this mechanism under TA, but also show that the core contribution of our method extends beyond TA in practice: across other merging rules, our method consistently reduces merging time while maintaining optimal or near-optimal accuracy. **This is consistent with your second suggestion: the current theoretical result should be scoped to TA, while the results on other rules should be presented as empirical extensions rather than as formally covered by Theorem 4.2.**
> > >
> > > ----
> > > > `Q2`
> > >
> > > A2:The core contribution of our work is that CircuMerge substantially improves merging efficiency, reducing the overall merging time by at least 44% compared to state-of-the-art approaches, while achieving optimal or near-optimal accuracy across different settings. **Thank you for your suggestion. We will revise the abstract in accordance with the aforementioned content to make the expression more precise.**
> > >
> > > ----
> > > We hope that these responses and revisions address your concerns. If so, we would greatly appreciate it if you would reconsider the score accordingly.

---

### Decision · Program_Chairs · 2026-04-30

**Decision:**

Accept (regular)

**Comment:**

This paper proposes CircuMerge, a sketch-based framework for efficient zero-shot LoRA adapter merging using shared circulant transforms. Reviewers generally found the work well-motivated, with an interesting theoretical insight linking effective rank to merging quality, providing a principled foundation for the method. The empirical results show competitive accuracy in average, and the rebuttal clarified the scope of the theory (now limited to TA), improved claims, and added supporting analyses. However, concerns remain the relatively marginal performance gains noted by reviewer, as well as questions about the practical importance of merge-time efficiency in some scenarios. Overall, the paper is technically sound and interesting, though its impact may be somewhat limited.